



# Correction of motion influence for nacelle based lidar systems on floating wind turbines

Moritz Gräfe[1], Vasilis Pettas[1], Julia Gottschall[2], and Po Wen Cheng[1]

[1]University of Stuttgart, Stuttgart Wind Energy (SWE), Allmandring 5b, 70569 Stuttgart, Germany
[2]Fraunhofer Institute for Wind Energy Systems IWES, Am Seedeich 45, 27572 Bremerhaven, Germany

**Correspondence:** Moritz Gräfe (graefe@ifb.uni-stuttgart.de)

**Abstract.** Inflow wind field measurements from nacelle based lidar systems offer great potential for different applications including turbine control, load validation and power performance measurements. On floating wind turbines nacelle based lidar measurements are affected by the dynamic behaviour of the floating foundations. Therefore, effects on lidar wind speed measurements induced by floater dynamics must be well understood. In this work we investigate the influence of floater motions

on wind speed measurements from forward looking nacelle based lidar systems mounted on floating offshore wind turbines (FOWT) and suggest approaches for the correction of motion induced effects. We use an analytical model, employing the GUM methodology and a numerical lidar simulation for the quantification of uncertainties. It is found that the uncertainty of lidar wind speed estimates is mainly caused by fore-aft motion of the lidar, resulting from the pitch displacement of the floater. Therefore, the uncertainty is heavily dependent on the amplitude and the frequency of the pitch motion. The bias of 10 min

mean wind speed estimates is mainly influenced by the mean pitch angle of the floater and the pitch amplitude.

Further, we discuss the need for motion compensation for different applications of lidar inflow measurements on FOWT and introduce two approaches for the correction of motion induced effects in lidar wind speed measurements. We correct motion induced biases in time averaged lidar wind speed measurements with a model based approach employing the developed analytical model for uncertainty and bias quantification. Testing of the approach with simulated dynamics from two different FOWT

concepts shows good results with remaining mean errors below 0.1ms[-1]. For the correction of motion induced fluctuation in instantaneous measurements we use a frequency filter to correct fluctuations caused by floater pitch motions for instantaneous measurements. The performance of the correction approach is dependent on the pitch period and amplitude of the FOWT design.

## 1 Introduction

With many countries worldwide having ambitious targets for FOWT installations and a pipeline of upcoming projects, the installed capacity of FOWT is expected to increase exponentially in the coming decade. Forecasts expect the global installed capacity of floating wind to reach 16.5 GW by 2030 (GWEC 2022). While FOWT offer the potential to exploit wind resources in deep waters, new challenges occur due to the dynamic behaviour of floating support structures. One of these challenges is the reliable measurement of wind resources and FOWT inflow conditions.





For wind measurements in deep waters typically met masts are not applicable due to high installation costs. As an alternative, floating lidar concepts have been developed and are already used in industry projects as well as research applications. A comprehensive overview of the technology and challenges can be found in Gottschall et al. (2017). For the measurement of inflow conditions to individual turbines the use of forward looking nacelle based lidar systems is advantageous. Potential use cases for nacelle based lidar systems include turbine control, power performance monitoring and load monitoring.

In Wagner et al. (2014) power curve measurements were performed with a nacelle based pulsed lidar system on a multi megawatt turbine and compared to measurements of a met mast mounted cup anemometer. The power curves obtained with lidar and cup measurements in general showed good agreement. It was found that the lidar power curve showed less scatter since the lidar measured wind speed is better correlated to the rotor effective wind speed of the wind turbine. Özinan et al. (2022) published a power curve assessment campaign with a nacelle based lidar on a 2MW FOWT. In this study the lidar

measurements were compared to a nacelle mounted sonic anemometer. While a good agreement between both wind speed references were observed, the power curves with lidar wind speed measurements showed higher scatter compared to the nacelle cup-anemometer measurements. This can be attributed to the influence of floater dynamics and the corresponding shift of measurement position on lidar measurements.

     In Conti et al. (2020) load simulations have been performed using lidar estimated wind field characteristics (WFC). Predicted

loads have been compared to load measurements at the turbine. In Conti et al. (2021) a methodology for combining nacelle based lidar measurements with constrained wind field reconstruction techniques has been proposed to improve accuracy of load assessments for fixed bottom wind turbines. In this study the lidar simulation framework ViConDAR Pettas et al. (2020) has been used for the realistic simulation of lidar measurements.

     Another application of nacelle based lidar systems is the use for different lidar assisted wind turbine control strategies. These

control strategies aim to use knowledge about the approaching wind field to optimize the operation of the turbine. Investigated concepts include collective and individual pitch control (see e.g.Bossanyi et al. (2014)), yaw control (see e.g. Fleming et al. (2014)) and speed control (see e.g. Schlipf et al. (2013)).

     Although different applications require wind speed measurements in different temporal resolutions (e.g. 1 Hz for turbine control versus 10min average for performance measurements), for all above mentioned applications the quantification of un-

certainties and biases in lidar measurements is essential. The need for uncertainty quantification and the development of suitable tools and models has also been highlighted as an imprtant step towards the broad application of nacelle based lidar systems by Clifton et al. (2018).

     While above mentioned studies have mainly investigated the use of nacelle based lidar systems for the measurement of inflow conditions of onshore or bottom fixed offshore wind turbines, little experience exists for the use on FOWT. Since the

floating dynamics of the FOWT causes translational and rotational displacement of nacelle mounted lidars, it can be expected that these dynamics affect the measurements. Therefore, it is necessary to investigate motion induced effects and to evaluate the need for motion correction.

     The quantification of uncertainties and correction of motion influence has in general already been approached by several authors for both, floating and fixed lidar systems. In Gottschall et al. (2014) motion induced effects on buoy based lidar





systems were investigated following a simulation based and experimental approach. While mean wind speed measurements showed little deviation from fixed reference measurements, the authors found systematically increased turbulence intensity (TI) measurements. Bischoff et al. (2022) introduced a simulation based approach for the uncertainty estimation of buoy based floating lidar systems under different met-ocean conditions. In Meyer and Gottschall (2022) a different, analytical, approach is followed for the investigation of uncertainties of nacelle based lidar measurements. In this work the methodology proposed by the guide for the expression of uncertainty in measurements (GUM) (JCGM 100:2008) is followed to estimate uncertainties due to line of sight variations in range, elevation angle, and azimuth angles. Kelberlau and Mann (2022) quantified the motion induced measurement errors for lidar buoys. Biases in the measurement were derived numerically and analytically using a mathematical model of the measurements under motion influence.

For nacelle based lidar systems on FOWT Gräfe et al. (2022) demonstrated the influence of floater dynamics on lidar measurements. Main effects influencing the obtained lidar wind measurements are changing beam directions, changing position of focus points and superposition of translational velocities. Rotational displacements of the floater and the lidar system cause tilted beam directions compared to a fixed lidar sytem, which leads to changing line of sight (LOS) measurements. Changing beam directions also cause changing positions of focus points in space. Under the presence of vertical wind shear this leads to errors in the wind speed estimates. Finally, floater dynamics cause translational displacements of the lidar system in space which creates the superposition of additional velocity components on the lidar measurements. The influence of floater dynamics on lidar measurements was investigated with a numerical simulation approach for a 15MW spar type FOWT. Results showed an increase of mean absolute error between lidar estimated and true wind field rotor effective wind speed depending on the environmental conditions. An overestimation of mean rotor effective wind speed, due to spatially shifted focus points was observed.

Kelberlau et al. (2020) suggest a motion compensation approach for a buoy mounted vertical azimuth display (VAD) scanning lidar system based on measured motion time series. Here, motion data from a motion reference unit is used to calculate the contribution of the buoy motions on the LOS measurements and the current LOS geometry. Further, this information is considered in the wind field reconstruction process for each measurement. In Désert et al. (2021) motion induced contributions to LOS wind speed measurement variances is calculated based on 10 min mean motion data and used to correct the turbulence measured by the lidar. A similar approach is followed by Gutiérrez-Antuñano et al. (2018). Here amplitudes and periods of the pitch and roll motion of a lidar buoy in combination with information about the lidar configuration are used to estimate the motion induced variance of horizontal wind speed estimates.

In Salcedo-Bosch et al. (2022) and Salcedo-Bosch et al. (2021) the use of Kalman Filters for motion correction of 10 min statistics from buoy based lidar systems is investigated. Inertial measurement unit (IMU) signals from the buoy along with a turbulence model are used to model the true wind velocity vector and correct the motion corrupted lidar measurements.

## 1.1 Objectives

With the present work we aim to systematically analyse the motion induced effects in nacelle based lidar measurements on FOWT and provide methodologies for the correction of these effects. In short the objectives of this work are:





    – to quantify floater motion induced uncertainties and biases in nacelle based lidar measurements on FOWT

– to introduce correction methods for motion induced effects on lidar measurements on different time scales

## 1.2  Structure of the work

In section 2, the overall methodology of the study, the used tools and measurement data sets are introduced. In section 2.2 we introduce a newly developed analytical model for the estimation of uncertainties and biases in lidar measurements for floating wind turbines under consideration of floater dynamics in yaw, pitch, roll and heave degree of freedom (DOF). Since

the analytical model includes several simplifying assumptions, we use the numerical lidar simulation framework ViConDAR (Pettas et al. (2020), Gräfe et al. (2022)) to verify the findings from the analytical model. The simulation approach of this framework is shortly introduced in section 2.3.

    In section 3 we discuss the motion influence on nacelle based lidar measurents. Therefore, we first present findings from the measurement campaign. Second we present a parametric study based on analytical and numerical model which quantifies the

influence of individual floater DOF on the lidar measurements.

    In section 4 we first discuss the need of motion compensation for different applications of nacelle based lidar measurements. Following this discussion, in section 4.1 we introduce a model based correction approach for 10 min averaged lidar wind speed estimations. Here, the analytical model is used to calculate correction values based on dynamics input parameters. In section 4.2 we introduce a correction approach for instantaneous lidar wind speed estimates based on time series frequency filtering.

In section 5 we use the numerical lidar simulation framework as testing environment for both correction approaches and evaluate the performance of the proposed motion correction approaches. Here we use simulated dynamics of two FOWT, modelled in the aeroelastic simualtion code openFAST (Jonkman (2007)) as inputs for the lidar simulation. A final discussion on the performance and the applicability of the correction apporaches for different FOWT characteristics is given in section 6.

## 2  Methodology

The methodology covers the introduction of the numerical and analytical models for quantification of uncertainty and bias in lidar wind measurements on floating wind turbines. A parametric study is used for the quantification of motion induced effects. Based on the findings of the parametric study two approaches for the correction of time averaged and instantaneous lidar wind measurements are introduced. An overview of the used methodology and use of the analytical and numerical lidar measurement models is shown in figure 1. A dynamics parameter space, defining the range of rotational and translational displacements is

defined. Using the analytical model in combination with the GUM methodology as well as the numerical model in combination with statistical metrics, uncertainties and biases in nacelle based lidar measurements are quantified. Using this quantification of uncertainties and biases, two correction approaches are introduced.





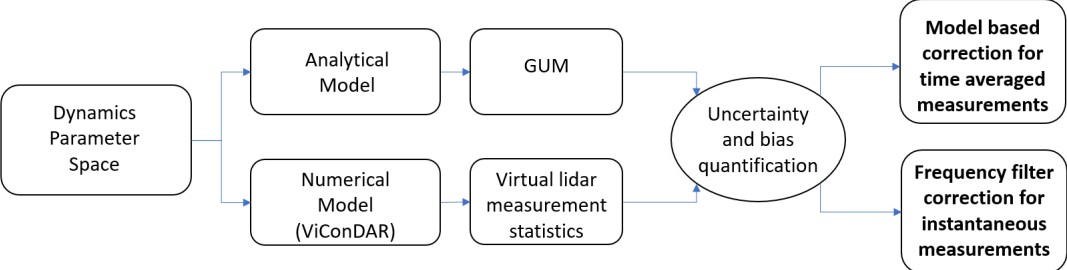

**Figure 1.** Utilization of analytical and numerical lidar measurement models.

## 2.1 Measurement campaign

The data set analysed for this study contains data from a forward looking lidar mounted on the nacelle of the FLOATGEN
demonstrator (BWIdeol (2019)) floating wind turbine. The floating substructure is a barge type floater employing BWIdeols
damping pool design. It is installed on the SEMREV test site (ECN (2017)) located near the atlantic coast of Brittany. The
wind turbine employed for this demonstrator is a 2MW Vestas V80 turbine with a rotor diameter of 90m and a hub height of
60m. Further details on the measurement campaign can be found in Özinan et al. (2022).

The measurement campaign employed a WindIris TC lidar (Vaisala (2022)). This lidar system is a 4-beam pulsed lidar with
a 1 Hz sampling frequency for the total pattern. LOS velocities are measured at ten distances from 50m to 200m. Furthermore,
the lidar system is equipped with an IMU, providing the rotational displacement in pitch and roll direction of the lidar system.
The wind field reconstruction procedure followed in this study is based on the approach from Schlipf et al. (2020). In this
method the horizontal wind speed components $u$ and $v$ are reconstructed using LOS measurements from all four beams. For the
rotational displacement measurements from the lidar IMU, it has been found that the measurements do heavily overestimate the
amplitudes of the displacements compared with other nacelle based sensors. Therefore, the inclination data used in this study
has been corrected using a linear relationship between the inclination measurements of the IMU and other nacelle mounted
inclinometers. The correction approach is discussed in Chen et al. (2022).

## 2.2 Analytical model

The analytical uncertainty model aims to provide estimates of uncertainty for lidar LOS wind speed measurements as well
as wind field characteristics reconstructed from LOS wind speed measurements. The uncertainty estimation according to the
GUM methodology requires an analytic description of the measurement. For the case of nacelle based lidar measurements on
a FOWT the model must contain the relevant dynamic behaviour of the FOWT, a description of the wind field and a model of
the measurement itself.

The dynamic behaviour of the FOWT is modelled considering four DOF, namely the rotational displacement in yaw, pitch
and roll direction as well the heave displacement of the floater (see figure 2). For simplicity, sway and surge displacement of



the floater are not considered in the model as individual DoF. The temporal behaviour of the considered DOFs is modelled by harmonic oscillations around a defined mean value:

$$a = A_a \cdot (sin\frac{2\pi}{T_a}t) + k_a \qquad (1)$$

where $A_a$ is the amplitude, $T_a$ is the period and $k_a$ is the mean value of the respective DOF. To transfer the modelled floater
dynamics on the dynamics of the lidar device mounted on the nacelle of the turbine, a rigid floater- tower assembly is assumed. Therefore, the rotational displacement and the translational heave displacement of the lidar device are equivalent to the displacement of the floater. The rotational displacements of the floater cause relevant translational displacements at the mounting position of the lidar. The translational displacements are modelled by defining the mounting position of the lidar, $[x_{FL}, y_{FL}, z_{FL}]$, in the floater coordinate system and the multiplication with a rotation matrix $R$.

$$\begin{bmatrix} x_{trans} \\ y_{trans} \\ z_{trans} \end{bmatrix} = R(\psi, \beta, \gamma) \cdot \begin{bmatrix} x_{FL} \\ y_{FL} \\ z_{FL} \end{bmatrix} + \begin{bmatrix} 0 \\ 0 \\ z_{heave} \end{bmatrix} \qquad (2)$$

The translational velocities at the lidar mounting position are found by calculation of the first time derivative of the translational displacement:

$$\boldsymbol{x_{vel}} = \frac{\Delta \boldsymbol{x_{trans}}}{\Delta t} \qquad (3)$$

The model parameters defining the dynamic behaviour of the FOWT are summarized in table 1.

**Table 1.** Considered FOWT dynamics parameters

| DOF | Amplitude | Period | mean |
| --- | --- | --- | --- |
| Yaw | $A_\psi[°]$ | $T_\psi[s]$ | 0 |
| Pitch | $A_\beta[°]$ | $T_\beta[s]$ | $k_\beta[°]$ |
| Roll | $A_\gamma[°]$ | $T_\gamma[s]$ | 0 |
| Heave | $A_{heave}[m]$ | $T_{heave}[s]$ | 0 |

In reality, the LOS measurement of a lidar is influenced by various atmospheric and technical parameters. For the analytical calculation of motion induced uncertainties a simplified model of the atmosphere and the lidar measurement is introduced. The horizontal wind speed is modelled using a power law profile given by:

$$V_h = V_{ref}(\frac{H}{H_{ref}})^\alpha \qquad (4)$$



where $V_{ref}$ is the reference wind speed, $H_{ref}$ is the reference height, $H$ is the height above ground and $\alpha$ is the vertical wind shear exponent. Horizontally, the wind field is assumed to be homogeneous. The $v$ and $w$ wind speed components are assumed to be zero. Having defined the wind field, we describe the LOS measurement $v_{los,i}$ as a function of the dynamic position of each focus point $x_I, y_I, z_I$ and the local wind vector components $u, v, w$:

$$v_{LOS_i} = f(x_I, y_I, z_I, u, v, w) \tag{5}$$

For the consideration of dynamic LOS position due to floater motion two coordinate systems are introduced. Following the notation of Schlipf (2016) the $I$ coordinate system is an earth fixed reference coordinate system and the $L$ coordinate system is the coordinate system of the lidar device. The coordinates of the lidar focus points are defined in terms of angles from the center line $\theta$ and angles around the center line $\phi$:

$$\begin{bmatrix} x_L \\ y_L \\ z_L \end{bmatrix} = \begin{bmatrix} \cos\theta \\ \sin\theta\cos\phi \\ \cos\theta\sin\phi \end{bmatrix} \tag{6}$$

Considering the rotational displacement of the lidar system due to the yaw, pitch and roll DOF of the floater, the coordinates of the rotated focus points in the earth fixed coordinate system are obtained by multiplication with a rotation matrix $R_{x,y,z}$:

$$\begin{bmatrix} x_I \\ y_I \\ z_I \end{bmatrix} = R(\psi, \beta, \gamma) \cdot \begin{bmatrix} x_L \\ y_L \\ z_L \end{bmatrix} \tag{7}$$

where $\psi, \beta, \gamma$ is the current roll, pitch, and yaw angle respectively and $[x_I, y_I, z_I]$ is the current position of the lidar focus points in earth fixed coordinates (see figure 2).

Finally, the LOS velocity is mathematically described by a projection of the local wind vector on the vector describing the line of sight. Probe volume averaging effects are not considered.

$$v_{los} = x_{In}u + y_{In}v + z_{In}w + (x_{In}x_{vel} + y_{In}y_{vel} + y_{In}z_{vel}) \tag{8}$$

where $x_{In,i}, y_{In,i}, z_{In,i}$ is the normalized LOS vector. The wind speed components $u_i, v_i, w_i$ are derived using equation 4. The translational velocities $[x_{vel}, y_{vel}, z_{vel}]$ are added to the local wind vector as additional velocity components. The dynamic measurement height $H$ is give by:

$$H = z_I + h_{lidar} + h_{heave} \tag{9}$$

where $z_I$ is the z-coordinate of the focus point, $h_{lidar}$ is the height of the lidar mounting position above mean sea level and $h_{heave}$ is the elevation due to the heave motion of the floating platform.





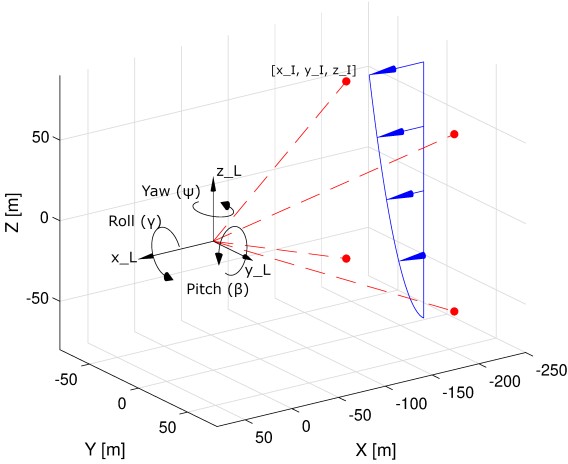

**Figure 2.** Lidar pattern and coordinate systems for the analytical model.

Considering the vertical wind profile, the wind direction $\varphi$ and assuming the vertical wind speed component $w$ to be zero yields:

$$v_{los} = V_{ref}(\frac{z_I + h_{lidar} + h_{heave}}{H_{ref}})^\alpha * (\sin\varphi * x_{In} + \cos\varphi * y_{In}) + (x_{In}x_{vel} + y_{In}y_{vel} + y_{In}z_{vel}) \tag{10}$$

Following the recommendation of IEC 61400-50-3:2022, the GUM methodology is applied for quantification of uncertainty in $v_{los}$ as function of input parameters given in equation 10. Further, this uncertainty is propagated through a wind field reconstruction algorithm to derive uncertainties of the reconstructed u-component of the wind vector. Details on the derivation of uncertainty are described in appendix A. The analytical model is also used to quantify bias in lidar estimated u-component

wind speed as function of dynamic input parameters. The derivation of bias is given in appendix A0.3. The model is made available as an open source tool by the chair for wind energy at University of Stuttgart (SWE) on *"https://github.com/SWE-UniStuttgart/FLIDU"*.

### 2.3   Numerical model

The analytical model for the estimation of uncertainty and biases introduced in chapter 2.2 includes several simplifying as-

sumptions about the wind field and the measurement. Particularly, it assumes a horizontally homogeneous wind field, does not account for turbulent effects and models the lidar measurement as a single point measurement. Therefore, a more sophisticated numerical model is employed to verify the analytical uncertainty and bias estimation. ViConDAR is an open source numerical framework for the simulation of lidar measurements in turbulent wind fields and the use of simulated measurements as constraints in synthetic wind field generation. Details on ViConDAR can be found in Pettas et al. (2020). ViConDAR has

been adapted for consideration of floating dynamics of the lidar system in 6 DOF. In this study we use ViConDAR to simulate lidar measurements under the same dynamic input quantities as in the analytical model and derive uncertainties and biases of



simulated measurements and reconstructed wind speed. ViConDAR requires the input of the rotational displacements (yaw, pitch, roll), the translational displacements in x,y,z direction as well as the translational velocities in x,y,z direction.

Similar to the analytical model the position of lidar focus points after rotational displacement is obtained by multiplication of the LOS vectors with a rotation matrix $R(\psi, \beta, \gamma)$, where $[x_I, y_I, z_I]$ is the current position of the lidar focus points in space. The LOS measurements of the individual beams are modelled as a projection of the wind vector $[u, v, w]$ on the current LOS vector $[x_I, x_I, x_I]$:

$$v_{los,i} = \int_{-\infty}^{\infty} (x_I u_i + y_I v_i + z_I w_i) f(a) da + (x_{vel} x_I + y_{vel} y_I + z_{vel} z_I) \tag{11}$$

where $f(a)$ is the range weighting function and $a$ is the measurement distance. In the simulation, the range weighting function
is represented by a definable length of the range gate and discretized by a number of points along the beam. The translational velocities of the lidar in $x, y, z$ direction create additional velocity components in the LOS measurements.

The translational displacement of the lidar system does not affect the angular beam directions but moves the lidar system in space. After consideration of translational displacements the positions of focus points are given by:

$$\begin{bmatrix} x_P \\ y_P \\ z_P \end{bmatrix} = \begin{bmatrix} x_I \\ y_I \\ z_I \end{bmatrix} + \begin{bmatrix} x_{trans} \\ y_{trans} \\ z_{trans} \end{bmatrix} \tag{12}$$

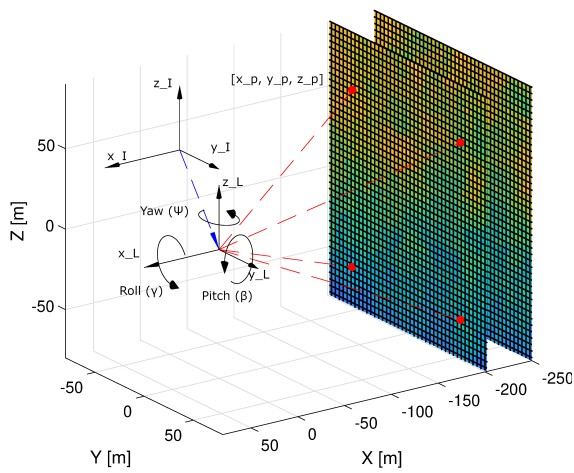

**Figure 3.** Lidar pattern and coordinate systems for the numerical model.

where $x_{trans}, y_{trans}, z_{trans}$ include the heave, surge and sway DOFs of the floating platform. In the simulation this is reflected by deriving the wind vector in equation 11 for position $[x_p, y_p, z_p]$. The x- coordinate of the focus points in space is





converted to the time coordinate of the synthetic turbulence box using Taylors frozen turbulence hypothesis. The wind vector is then sampled from the closest point to the $[x, y, z]$ grid of the synthetic turbulence box (see figure 3).

Synthetic turbulent wind fields with desired characteristics are created using a turbulence generator. In this work the open source turbulence generator TurbSim (Jonkman (2014)), which is based on employing the Veers method Veers et al. (1998) for turbulence modelling is used. The u-component wind speed is reconstructed following the same approach as the reconstruction procedure for the analytical model described in appendix A0.2.

## 3 Motion influence in nacelle based lidar measurements

### 3.1 Findings from measurement campaign

Figure 4 shows the 10 minute IMU mean pitch angle per wind speed including the standard deviation of 10 minute mean pitch angle per wind speed bin for the floating example data set. As expected the mean pitch angle data shows a high dependency on wind speed and related thrust force. However, the magnitude of mean pitch angles is in the region below 1 deg which is rather low compared to other floater types (e.g. WindCrete FOWT concept, Mahfouz et al. (2021)). Therefore, no large errors in the mean wind speed estimates due to the mean pitch angles are expected for this floater type.

Additionally, the standard deviation within each 10min interval is shown per wind speed bin. Standard deviations in pitch angle are increasing with the wind speed and the associated wave excitation of the floater. For this floater type, the pitch motion is mainly caused by hydrodynamic forces, and is large compared to other floater types (e.g. spar type floaters). It can be expected that the pitch motion and related translational velocities of the nacelle will cause significant fluctuations in the lidar measurements.

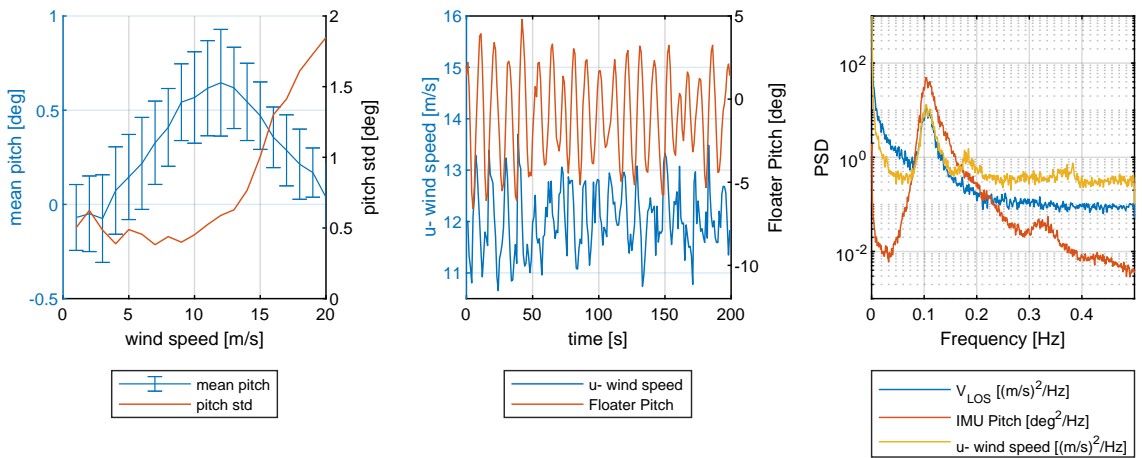

**Figure 4.** Left: Mean and standard deviation of floater pitch angle over 10min mean wind speed bins. Center: Time series of u- component wind speed and pitch angle. Right: PSD of floater pitch angle, LOS velocity and reconstructed u- component wind speed.





This is confirmed by the time series example (center) in figure 4, which shows the lidar reconstructed wind speed estimate as well the measured pitch inclination signal. A strong correlation between the pitch angle of the floater and the lidar wind speed can be observed.

Finally, the power spectral density (PSD) of the IMU pitch signal, of one LOS velocity signal and the reconstructed u-component of the wind speed is shown. The PSD of the lidar IMU inclination signal shows two peaks. The first peak corresponds to the pitch natural frequency of the floater at around 0.1 Hz, while the second peak is the tower natural bending frequency at around 1Hz. The PSD of the LOS velocities of the individual beams as well as the reconstructed u-component of the wind speed show peaks at the floater pitch frequency. For this example a clear correlation between the lidar pitch signal and the reconstructed u-component wind speed can be observed. This analysis shows that the rotational DOF of the floater, in particular the pitch motion, is strongly influencing the LOS measurements as well as the reconstructed wind speed of the nacelle mounted lidar. These findings suggest, that the motion influence on the lidar measurements needs to be further investigated and quantified.

## 3.2 Parametric study for uncertainty and bias quantification

In this section the findings from uncertainty and bias estimation of lidar measurements under motion influence are presented for both- the analytical and the numerical approach. The lidar configuration investigated in the numerical and analytic uncertainty estimation follow the one which is used in the real measurement campaign. This configuration represents a fixed beam lidar system with 4 beams, arranged in a rectangular pattern. The opening angle (angle to the center line) of all four beams is set to $\theta = 19.8°$ and the angle around the center line is set to $\phi = 39.6°, 140.4°, -39.6°, -140.4°$. The range of the lidar is assumed to be 200m.

The LOS measurements from all 4 beams are taken sequentially with a delay of $T_{meas} = 0.25\ s$. For this pattern, the duration of a full scan is 1 s. The resulting scanning pattern is visualized in figure 3. For both the analytical model as well as the numerical model, a hub height which is also the installation height of the lidar, of 100m and a rotor radius of 75m is assumed.

In a first analysis the influence of individual DOF on the measured LOS velocities is examined with the help of the analytical model. Figure 5 shows the LOS velocity per beam as function of displacement for the yaw, pitch, roll and heave displacement for three values of the vertical shear exponent $\alpha$. Additionally, the reconstructed u- component of the wind field is shown as function of each DOF.

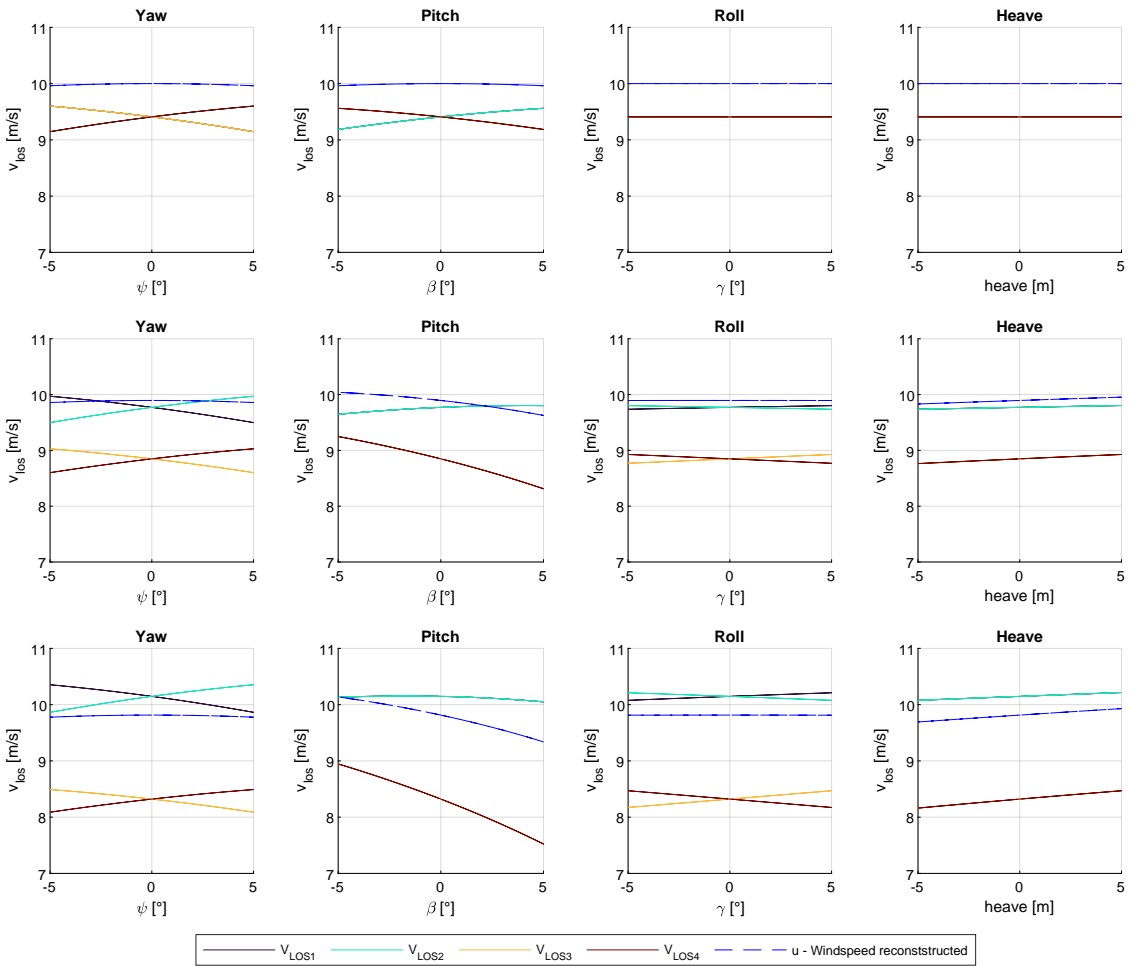

**Figure 5.** Modelled LOS velocities and reconstructed u- component wind speed as function of yaw, pitch, roll and heave displacement for all four beams. Reference wind speed: 10 ms$^{-1}$. Top: $\alpha = 0$. Center: $\alpha = 0.1$: Bottom: $\alpha = 0.2$. $\alpha$ is the vertical shear exponent.

For a vertical shear exponent of $\alpha = 0$, the individual LOS velocities differ for changing pitch and yaw angles. However, due to the symmetric beam pattern the reconstructed wind speed shows no significant fluctuation over the respective DOF. Under the presence of a vertical wind shear profile this behaviour changes. The yaw and roll displacement causes fluctuations in the

270 LOS velocities. Again the reconstructed wind speed stays constant due to the symmetric beam pattern.

For the pitch DOF, the upper and lower beams LOS velocities have a different characteristics. Therefore, the reconstructed wind speed is fluctuating. It is also important to note that the relationship between reconstructed wind speed and pitch angle is non-linear. This non linear relationship can cause bias in averaged lidar wind speed estimates. Heave displacement in combination with non-linear vertical wind shear profile causes fluctuation in the reconstructed wind speed. However, the relationship





between displacement and reconstructed wind speed is almost linear which indicates that no significant bias is introduced by the heave displacement.

Figure 6 illustrates the dependency of measured LOS velocities and reconstructed wind speed on translational velocities of the lidar device. It can be seen, that the translational velocity components are directly projected on the LOS direction of the beam, and thus cause significant changes in the measured LOS velocities. For the translational displacement in x- direction,

this directly translates into changing reconstructed wind speeds. For the y- and z- direction the reconstructed u- component wind speed remains constant since the effects on the left / right and upper / lower beams compensate each other. However, it is important to note, that the relationship is strictly linear, meaning that zero mean fluctuations in translational velocities will cause zero mean fluctuations in LOS velocities and thus not cause any bias in the LOS velocity and reconstructed u- component of the wind speed.

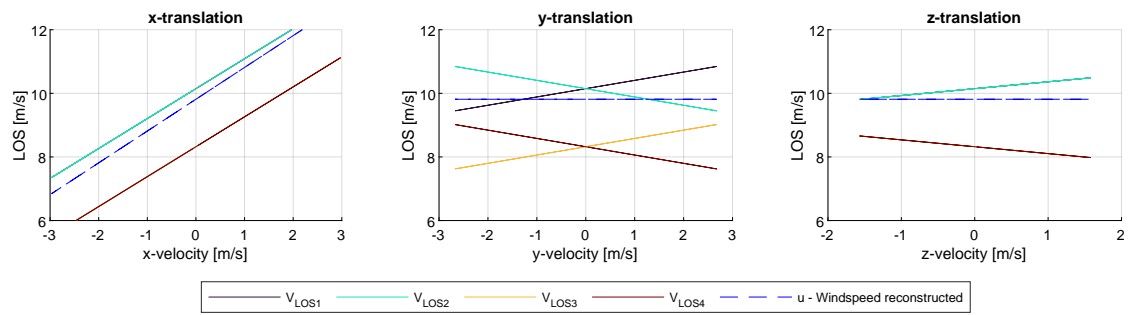

**Figure 6.** Modelled LOS velocities as function of translational nacelle velocities in x-, y- and z- direction for all four beams. Reference wind speed: 10 ms$^{-1}$.

This analysis shows that the most relevant DOF for the influence of nacelle based lidar wind measurements is the pitch motion. Under the presence of a vertical wind shear profile, the rotational pitch displacement causes a significant variation of reconstructed wind speed. This variation is non linear as function of pitch displacement, indicating that the displacement could introduce bias in the measurement. The rotational pitch motion also causes translational velocities of the nacelle in x-direction. These translational velocities cause fluctuations of the reconstructed wind speed. Consequently, we focus on the analysis of the

influence of the pitch DOF in combination with the present vertical shear profile in the following sections.

### 3.3 Uncertainty quantification

In this section we quantify motion induced uncertainties in reconstructed lidar wind speed estimates as function of dynamic input parameters using the analytical as well as the numerical model. For the analytical model the wind field is only defined by the inflow wind speed of 10 ms$^{-1}$ and the vertical shear exponent $\alpha$. The dynamics parameters are summarized in table 2.

Figure 7 first row shows the u-component wind speed uncertainty estimates from the analytical model as functions of shear exponent, mean pitch angle and the pitch amplitude.





**Table 2.** Dynamics parameters for uncertainty quantification.

| Parameter / case | a) | b) | c) | d) |
|---|---|---|---|---|
| Pitch Amplitude [deg] | $0:1:5$ | $0:1:5$ | 2 | 2 |
| Mean Pitch [deg] | 0 | -5 : 1 : 0 | 0 | -5 : 1 : 0 |
| Pitch Period [s] | 30 | 30 | $10:5:50$ | $10:5:50$ |

The resulting uncertainty of the reconstructed u- wind speed component is mainly dependent on the pitch amplitude, which determines the magnitude of different effects. With a given pitch period of 30s, the pitch amplitude determines the magnitude of the translational velocity in fore-aft direction. In this case, this is the main source of uncertainty in the reconstructed wind 300 speed. Additionally, the pitch amplitude determines the uncertainty resulting from changed beam direction and shifted focus points. The overall uncertainty of the u- wind speed component is found to be in the region of up to 20% of the reference wind speed. No significant influence of the inflow shear exponent and the mean pitch angle on the overall uncertainty estimate of the u-component wind speed component can be observed.

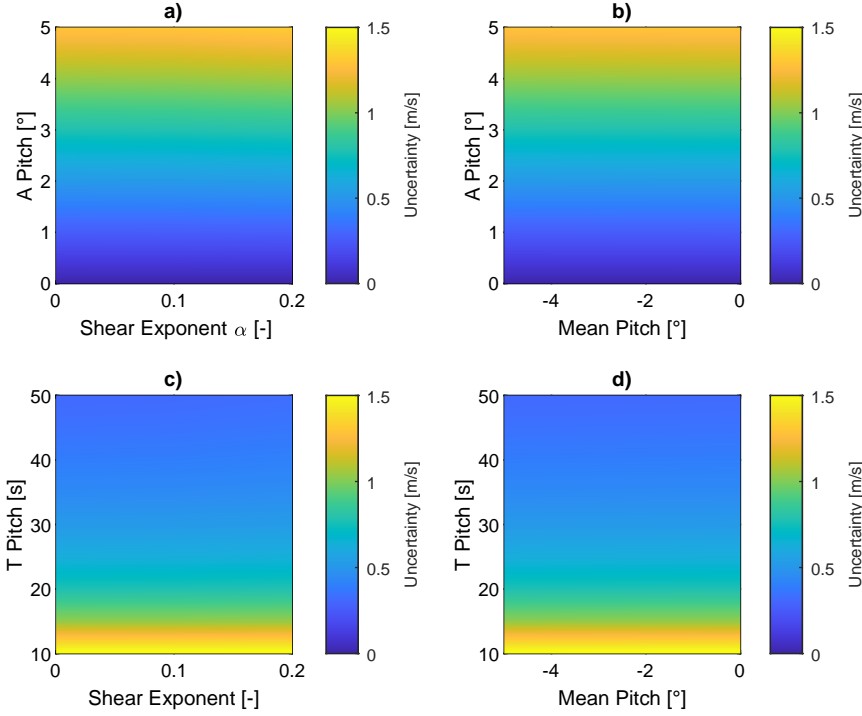

**Figure 7.** Uncertainty estimation of u-component wind speed. Fixed Parameters: $U_{ref}$: 10ms$^{-1}$, $A_\psi = 0$ deg, $A_\gamma = 0$ deg, $A_{heave} = 0$.



In a next step, the influence of the frequency of rotational pitch movement was investigated. Figure 7 second row shows the uncertainty of u-component wind speed as function of shear exponent, mean pitch angle and period of floater pitch movement. The corresponding parameter space is summarized in table 2. Results show a strong dependency on the period of the rotational pitch movement of the floater. This parameter determines the magnitude of the translational velocities at the lidar mounting position. Therefore, short periods of rotational movement in general cause high uncertainty in u-component wind estimates. No significant influence of the inflow shear exponent and the floater mean pitch angle can be observed.

A verification of the analytical results is done employing the numerical model. Turbulent synthetic wind fields used in the simulations are generated for a mean wind speed of 10 m/s and a turbulence intensity of 6%. 10 random turbulence realizations are generated for each wind condition and the presented results are averaged over all random realizations. Additional wind field parameters are summarized in table 3.

**Table 3.** Wind field parameters.

| Parameter | Value |
|---|---|
| Wind speed [ms$^{-1}$] | 10 |
| Turbulence Intensity[%] | 6 |
| Surface Roughness [m] | 0.03 |
| Shear exponent [-] | 0.0 : 0.1 : 0.2 |
| Spatial Grid Resolution [m] | 4 |
| Gridsize [m] | 180x180 |
| Timestep [s] | 0.1 |
| Usable Time [s] | 600 |
| Seeds | 10 |

Figure 8 shows a comparison of the analytical and numerical uncertainty estimation for selected dynamic input parameters. For the numerical model, uncertainty is represented by mean absolute error (MAE) metric. The MAE is the mean of the absolute error between the time series of lidar estimated wind speed and the reference time series of the full input wind field. The MEA represents the instantaneous error between reconstructed wind speed and the input wind field. Therefore, it contains the fluctuations resulting from assumed FOWT dynamics. This metric can be compared qualitatively and quantitatively to the uncertainty results from the analytic model.

In general, the pattern of the numerical results follows the estimation of the analytical uncertainty model. The MAE is heavily dependent on the pitch amplitude and period, which is determining the magnitude of the fore-aft motion of the lidar system. Shear exponent and mean pitch angle variation show no significant effect on the MAE. Quantitatively, both models show similar magnitudes of uncertainty and MAE values respectively. It can be seen that there is a linear relationship between





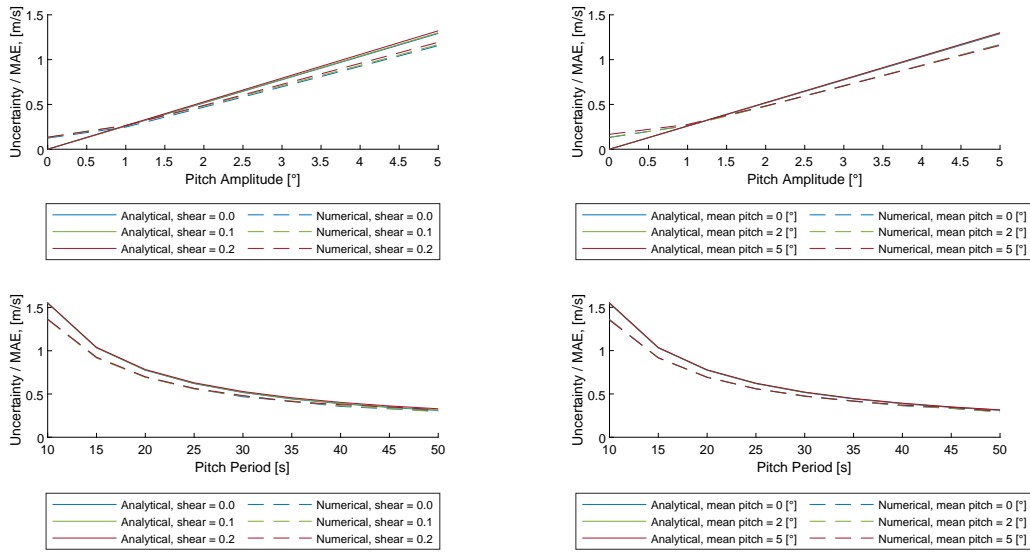

**Figure 8.** Comparison between analytical and numerical uncertainty / MAE estimation. Fixed Parameters: $U_{ref}$: 10ms$^{-1}$, $A_\psi = 0$ deg, $A_\gamma = 0$ deg, $A_{heave} = 0$, $\alpha = 0.1$.

uncertainty in u- wind speed estimate and the amplitude of the pitch DOF of the floater. A nonlinear relationship between
uncertainty in u- wind speed estimate and pitch period with lower uncertainties for increasing pitch periods can be observed.

### 3.4 Bias quantification

In this section we quantify the motion induced bias in reconstructed lidar wind speed estimates as function of dynamic input parameters using the analytical as well as the numerical model. The input wind fields for the numerical model as well as the dynamics parameter space is the same as previously used for the uncertainty quantification.

Figure 9, first row, shows the estimation of bias in the reconstructed u-component wind speed from the analytical model. As a function of vertical shear exponent and pitch amplitude, results show a negative bias in the reconstructed u-component wind speed which is increasing for higher shear exponents. For increasing pitch amplitudes a slightly negative trend in the wind speed estimation can be observed, which is more pronounced for high vertical shear conditions. This is caused by the combined effect of a nonlinear vertical wind shear profile and the nonlinear relationship between beam directions and measured
LOS velocity.

Non-zero mean pitch angles result in general in upwards or downwards shifted focus points. For negative mean pitch angles (upwards shifted focus points) and the assumption of a power law wind profile this results in a positive bias in the LOS measurements. Depending on the magnitude of vertical shear this effect exceeds the negative effect from the pitch motion and




leads to an overestimation of wind speed. Consequently, the overestimation is most pronounced for high negative mean pitch
and low pitch amplitudes.

For the relationship between vertical shear, pitch amplitude and resulting bias, it can be seen that the calculated bias is not zero, even in the presence of zero pitch amplitude. This bias is not introduced by any dynamics but is a result of the definition of the reference value used for bias calculation. As detailed in appendix A, the reference for bias calculation is defined as the average wind speed over the rotor plane. Depending on the present shear profile, the assumed rotor size and lidar pattern, this
reference value is different from the lidar wind speed estimate. Therefore, only the change of bias over the varying dynamics parameters can be attributed to the dynamics influence.

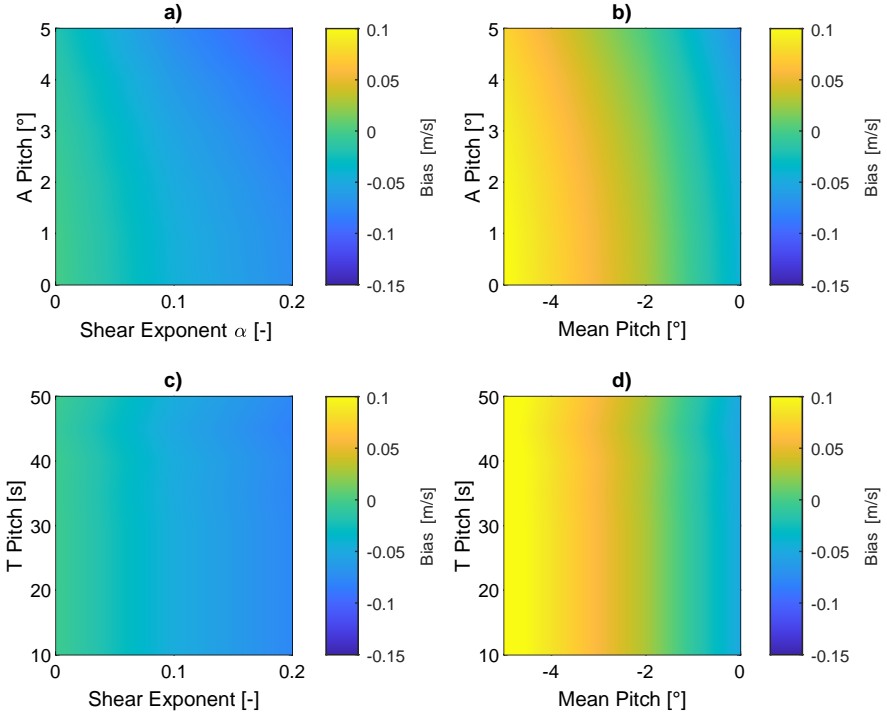

**Figure 9.** Bias of u-component wind speed. Parameter according to table 2. Fixed Parameters: $U_{ref}$: 10ms$^{-1}$, $A_\beta$ = 2 deg ,$A_\psi$ = 0 deg, $A_\gamma$ = 0 deg, $A_{heave}$ = 0 m.

Figure 9 second row shows the bias of u-component wind speed as function of shear exponent, mean pitch angle and pitch period. No significant dependency of wind speed bias to the frequency of the floater pitch motion can be observed.

The analytical model results are verified using the numerical model. Figure 10 shows a comparison between the numerical
and analytical bias estimation. For the numerical model the bias is expressed in terms of the mean error (ME) which is the error between the mean of lidar estimated time series and the mean of full wind field time series over the full simulation length of 600 seconds. The ME can be compared to the bias metric from the analytical model. As for the uncertainty estimation, the ME




follows the pattern of the analytical bias estimation. Quantitatively, the results of both models show good agreement. Observed deviations between the model results are in the region below $0.05\text{ms}^{-1}$ for the given reference wind speed of $10\text{ms}^{-1}$.

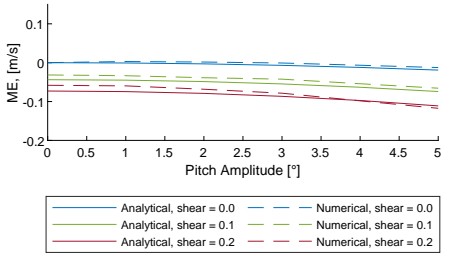
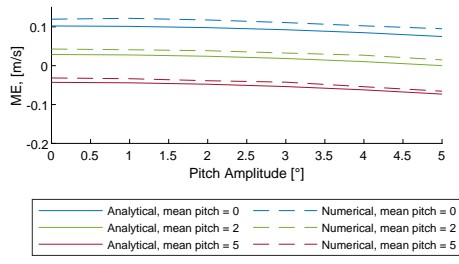

**Figure 10.** Comparison of bias wind speed estimation from analytical and numerical model. Fixed Parameters: $U_{ref}$: $10\text{ms}^{-1}$, $A_\psi = 0$ deg, $A_\gamma = 0$ deg, $A_{heave} = 0$m, $\alpha = 0.1$.

## 4   Correction Approaches

The results of the numerical and analytic uncertainty estimation have identified three main dynamic effects influencing the measurements of nacelle based lidars on floating wind turbines:

- translational velocity components caused by the fore-aft movement of the nacelle due to floater pitch motion cause fluctuation of horizontal wind speed estimates.

- rotational oscillations of the beam direction due to floater pitch motion cause an underestimation of horizontal wind speed components.

- negative mean pitch angles of the FOWT are causing upwards shifted focus point resulting in overestimation of wind speed.

It is important to note, that these effects have different orders of magnitude and are relevant at different time scales. Translational velocities cause fluctuations of the measurement time series which can be in the order of $1\ \text{ms}^{-1}$ and must be considered when using measurement time series data. Over- and under estimation of mean wind speeds due to rotational movement are one order of magnitude smaller and found to be in the region of $0.1\ \text{ms}^{-1}$. This is most relevant in cases where accurate estimates of mean wind speed is necessary.

Depending on the intended use of the measurement a correction of these effects can be necessary. Power performance measurement of wind turbines is a key application of lidar wind measurements. According to IEC 61400-12-1 the performance measurements should be performed by measuring the 10 minute average power output of the wind turbine as well as the 10 minute average inflow wind speed. A wind turbine power curve is then obtained by binning the wind speeds from $4\text{ms}^{-1}$ to $16\text{ms}^{-1}$ and plotting against the corresponding power outputs. While the standard procedure requires the use of met mast





mounted cup anemometers, the use of nacelle based lidar is an attractive alternative. However, the question arises if wind speed

measurements from nacelle mounted lidars introduce uncertainty or bias in the power performance measurements and if motion induced effects need to be corrected. The findings from section 3.4 show that mean wind speed estimates from nacelle based lidars can be significantly biased depending on the dynamic condition of the wind turbine. This suggests that a correction of 10 min mean wind speed estimated, used for power performance assessment will be necessary for the use nacelle based lidar systems on floating wind turbines under specific dynamic conditions.

The use of lidar inflow measurement for turbine load validation has been investigated by several studies using different methods. In Conti et al. (2020) 10min statistics of lidar estimated WFC are used for the parameterization of synthetic turbulent wind fields for aeroelastic turbine simulations and evaluation of loads. It was found that uncertainties in the lidar estimated mean wind speed and the lidar estimated turbulence intensity used for parameterization of synthetic wind fields is the main source of uncertainty in predicted loads. It can be expected that the dynamics induced fluctuations in lidar wind speed measurements

which are indicated by high measurement uncertainties will impact the lidar estimated turbulence statistics. Therefore, this will likely also influence load estimations. In Dimitrov et al. (2019), besides lidar estimated wind field statistics for parameterization, measured lidar wind speed time series are used to constrain synthetic wind fields for load validation studies. Also in this method it can be expected, that motion induced fluctuations in the wind speed measurements will significantly influence the resulting wind field and load estimates. This suggests that the correction of instantanous errors in lidar wind speed measurements

is necessary for the use in load validation studies.

Another application of nacelle based lidar systems on FOWT is the use of lidar wind speed measurements for turbine pitch control. Here the wind speed information of the inflow wind field is used as an input to a additional control loop which aims to compensate changes in wind speed, e.g. through gusts, by changing the rotor blade pitch angles in order to maintain the the rotor speed. This control approach can significantly reduce platform motions and variations in rotor speed due to disturbance

in the form of wind gusts (see e.g. Schlipf et al. (2015)). The application relies on instantaneous time series information of the inflow wind speed which contains the variation of wind speed on small time scales. Since these measurement contain motion induced fluctuations, correction of the lidar wind speed measurement time series data is necessary to avoid undesired effects on the pitch controller. In Schlipf et al. (2015) this is considered by using a model based wind field reconstruction algorithm which takes the instantaneous displacements and velocities of the lidar system into account to provide a motion compensated

wind speed estimate.

Above mentioned use cases show, that motion correction of measurements from nacelle based lidar systems is necessary while different time scales have to be considered. In this work we suggest a method for the correction of fore-aft motion induced fluctuation based on frequency filtering and a simple model based correction approach of the turbines mean pitch employing the analytical model.

## 405 4.1 Model based correction

As shown in section 5, the bias in the wind speed estimation is mainly caused by a non-zero mean pitch angle which causes upwards shifted measurement positions as well as oscillating beam directions caused by the floater pitch motion.



Instead of correcting lidar measurement time series based on the instantaneous turbine tilt angles, we suggest to correct the reconstructed wind speed with a model based approach. The analytical model introduced in section 2.2 is used to calculate the mean error in the reconstructed u-component wind speed estimation as a function of the wind speed $WS$, vertical shear exponent $\alpha$, turbine mean pitch $\beta_{mean}$ and pitch amplitude $A_{\beta}$.

$$v_{correction} = f(WS, \alpha, \beta_{mean}, A_{\beta}) \tag{13}$$

In this way a four dimensional look up table with correction values $v_{correction}$ is created for the parameter space given in table 4.

**Table 4.** Parameter space correction look up table.

| Parameter | Value |
|---|---|
| Wind speed [ms$^{-1}$] | $4 : 4 : 20$ |
| Shear [-] | $0.0 : 0.05 : 0.3$ |
| $\beta_{mean}[deg]$ | $0 : 0.1 : 5$ |
| A$_{\beta}[deg]$ | $0 : 0.1 : 5$ |

The remaining model parameters are set to fixed values of $T_{\beta} = 30s$, $A_{\psi} = 0$, $A_{\gamma} = 0$, $A_{heave} = 0$ as results are only evaluated for these values. It should be mentioned that the dimension of the created correction look up table can easily be extended to other model parameters if a significant sensitivity is found for a specific set up.

Assuming the mean pitch angle and the mean pitch amplitude is known for each 10 min period through inclination measurements and using the lidar estimated u-component wind speed and vertical shear exponent the correction value can be extracted from the look up table and subtracted from the lidar estimated u- component wind speed:

$$u_{corr.} = u_{rec} - v_{corr}(u_{rec}, \alpha_{rec}, \beta_{mean}, A_{\beta}) \tag{14}$$

## 4.2 Frequency Filtering

In this study we investigate the use of frequency filters for the correction of motion induced fluctuations. We suggest the application of a frequency filter on the time series of reconstructed u- component wind speed estimates. As shown in figure 4, the time series of LOS wind speed as well the reconstructed u- wind speed show peaks at the floater pitch frequency. The application of a frequency filter aims to correct the influence of floater pitch displacement and the resulting translational velocities in fore-aft direction, without introducing bias or error in the estimation of the real wind field properties.

For frequency filtering we use a bandstop filter characterized by three parameters. The stopband frequency range is given by a $width$ parameter, defining the upper and lower frequency limit of the stop band. The peak pitch frequency of the floater is used as the center frequency $f_{center}$ of the applied band stop filter, while the upper and lower bounds of the filter are



defined at $f_{pass,lower} = f_{center} - width/2$ and $f_{pass,upper} = f_{center} + width/2$. The filtering depth is defined by the stop band attenuation parameter, $depth$ in dB. The steepness of the filters transition region is defined by a steepness parameter.

The filter parameters applied in section 5 are optimized with a parametric study, evaluating the ME and the MEA between the corrected lidar wind speed and the full wind field reference. The goal of this optimization is to find the filter parameterisation
which is minimizing the MEA while not increasing the ME significantly. The filter has been implemented using the MATLAB (MATLAB (2020)) bandstop filtering function.

## 5  Results

We evaluate the correction approaches using the numerical lidar simulation framework ViConDAR. First lidar measurements are simulated using a prescribed dynamics parameter space with varying pitch amplitude, mean pitch angle and pitch period.
The parameter space defining the input dynamics is given in table 5. The assumed lidar set-up and turbine parameters are the same as in section 3.2. Turbulent synthetic wind fields are created according to parameters given in table 6. Second lidar measurements are simulated by coupling an aeroelastic simulation of specific FOWT models to ViConDAR. In this way the dynamics input response to wind and wave conditions are used as an input to the lidar simulation. Details on the coupling approach can be found in Gräfe et al. (2022). For this study we use two different FOWT models with different characteristics
in their input response to wind and wave conditions.

**Table 5.** Parameter space prescribed dynamics.

| Parameter | Value |
| --- | --- |
| Wind speed [ms$^{-1}$] | $4 : 4 : 20$ |
| Shear [-] | $0.0 : 0.1 : 0.2$ |
| $\beta_{mean}$ [deg] | 0, 3, 5 |
| $A_{\beta}[deg]$ | 0, 3, 5 |
| $Tp_{\beta}[s]$ | 10, 20, 30, 50 |

The first FOWT model employed for the simulation of FOWT dynamics is the WindCrete floater design concept (Mahfouz et al. (2021)) in combination with the International Energy Agency (IEA) 15 MW reference wind turbine (Gaertner et al. (2014)). WindCrete is a monolithic spar design with a draft of 155 m and a tower height of 129.5 m. The hub height of this FOWT concept is at 140m above sea level. The floater and the tower are designed as a single concrete member with an overall
mass of 3.665 x 10$^7$ kg. Three delta-shaped catenary mooring lines are employed for station keeping of the floater. The mooring lines have an overall length of 615 m and a mass per length of 561.25 kgm$^{-1}$. Details on the WindCrete design parameters and the dynamic behaviour can be found in Mahfouz et al. (2021).





The second FOWT model is the numerical model of the FLOATGEN FOWT demonstrator, introduced in section 2.1. Both FOWT are modelled in the open source aeroelastic simulation code OpenFAST (Jonkman (2007)).

**Table 6.** Wind field parameters.

| Parameter | Value |
|---|---|
| Wind speed [ms$^{-1}$] | 4 : 4 : 20 |
| Turbulence Intensity[%] | 6 |
| Surface Roughness [m] | 0.03 |
| Shear exponent [-] | 0.1, 0.2, 0.3 |
| Spatial Grid Resolution [m] | 5 |
| Timestep [s] | 0.05 |
| Usable Time [s] | 1200 |
| Seeds | 6 |

To minimize the influence of transient effects, the first 600s of each simulation are discarded. Simulations are created for three sets of wave conditions using a Jonswap wave spectrum with parameters given in table 7. For all testing cases the same lidar configuration as described in section 3.2 is used.

**Table 7.** Parameter space wave conditions.

| Parameter | Wave 1 | Wave 2 | Wave 3 |
|---|---|---|---|
| $H_s[m]$ | 1 | 2 | 5 |
| $T_p[s]$ | 6 | 9 | 11 |

### 5.1 Model based correction

The correction approach is first tested using the lidar simulation framework ViConDAR and prescribed dynamics inputs. IEC
IEC 61400-12-1:2016 suggests the verification of lidar wind measurements against reference wind speed measurements. The measurements from the remote sensing device and the reference sensor (e.g. a cup anemometer) shall be compared bin wise for wind speeds from 4ms$^{-1}$ to 16ms$^{-1}$. Following this approach, figure 11 and 12 show the relative uncertainty of a typical cup anemometer used for power performance measurements over a averaging period of 10 minutes as a reference. The reference values are extracted from IEC 61400-12-1:2016.
Figure 11, first row shows the resulting mean errors between the input wind field and the lidar estimates of the u- wind speed component without correction. It can be seen, that the relative ME is mainly influenced by the mean pitch angle in combination with the present vertical wind shear profile, while it is almost constant over wind speed. For a vertical shear exponent of 0.2



and a mean pitch angle of 3 deg the overestimation of wind speed is in the region of around 3% which exceeds the shown reference uncertainty.

Figure 11, second row shows the mean error for the same parameter space after correction. It can be seen, that the mean error can be reduced significantly and is below 1% of the wind speed over the entire range of wind speeds.

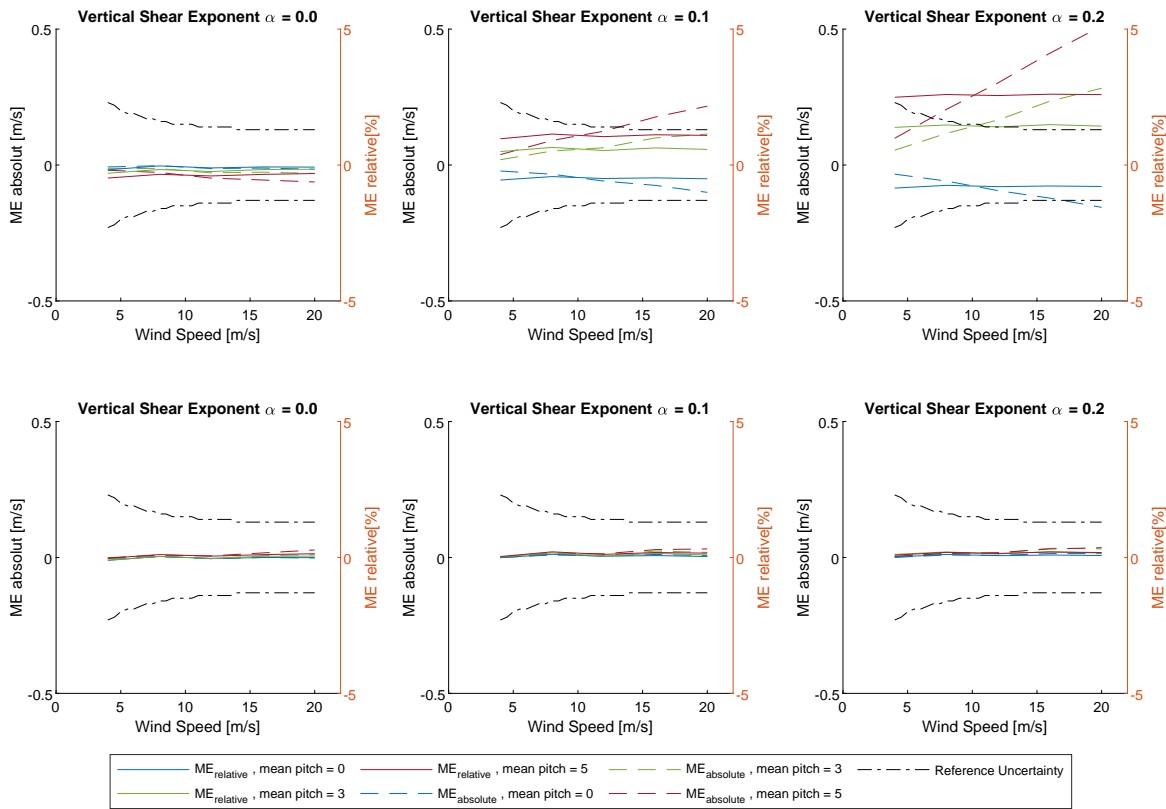

**Figure 11.** Absolute and relative ME of corrected u- wind speed component as function of mean wind speed for different mean pitch angles. Top: Uncorrected lidar estimates. Bottom: Corrected lidar estimates. Fixed Parameters: $A_\beta = 3$ deg, $A_\psi = 0$ deg, $A_\gamma = 0$ deg, $A_{heave} = 0$ m.

Similarly, figure 12 shows the mean error before (first row) and after correction (second row) for different pitch amplitudes. While the higher pitch amplitudes in combination with positive vertical shear exponents cause a negative mean error for the uncorrected u- wind speed estimations, the effect can be significantly reduced by the correction approach. Remaining errors

are in the region below 1% of the wind speed.

Figure 13 shows the model based correction results for simulated dynamics for the FLOATGEN (first row) and WindCrete (second row) FOWT. Before correction FLOATGEN shows negative mean errors in u-component wind speed, which increase with wind speed. This pattern is more pronounced under wind conditions with high vertical shear values. This behaviour can be explained by the dynamics characteristics of FLOATGEN. As a barge type floater, FLOATGEN shows insignificant mean



pitch angles for the given conditions. On the other hand, floater pitch amplitudes show relatively high values for the given wave condition. Consequently, the negative effect of pitch movement is predominant. The model based correction approach is able to reduce the ME significantly resulting in ME below 0.1 ms$^{-1}$.

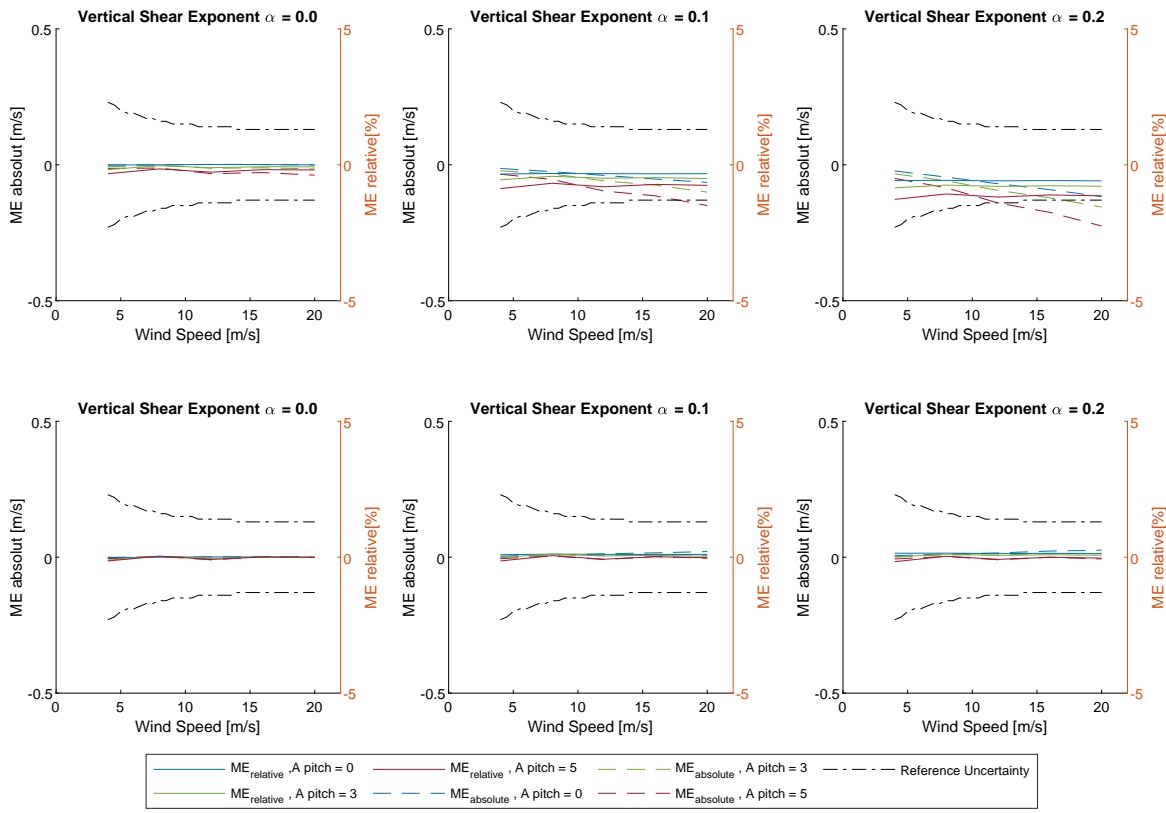

**Figure 12.** Absolute and relative ME of corrected u- wind speed component as function of mean wind speed for different mean pitch angles. Top: Uncorrected lidar estimates. Bottom: Corrected lidar estimates. Fixed Parameters: $\beta_{mean}$ = 0 deg, $A_\psi$ = 0 deg, $A_\gamma$ = 0 deg, $A_{heave}$ = 0 m.

The WindCrete FOWT model shows a different behaviour. The mean error of the u-component wind speed before correction increases with the wind speed up to the rated wind speed of the turbine and slightly decreases for above rated wind conditions.

The effect is more pronounced for high vertical shear conditions. The spar type floater only shows small pitch amplitudes, below one degree, for the given wind and wave conditions. In contrast to FLOATGEN, the mean pitch angle shows relatively high values, especially at rated wind speed. Consequently, the positive effect of upwards shifted lidar focus points is predominant. In this case the model based correction approach can reduce the mean error significantly over the full wind speed range. Remaining MEs are below 0.05ms$^{-1}$.

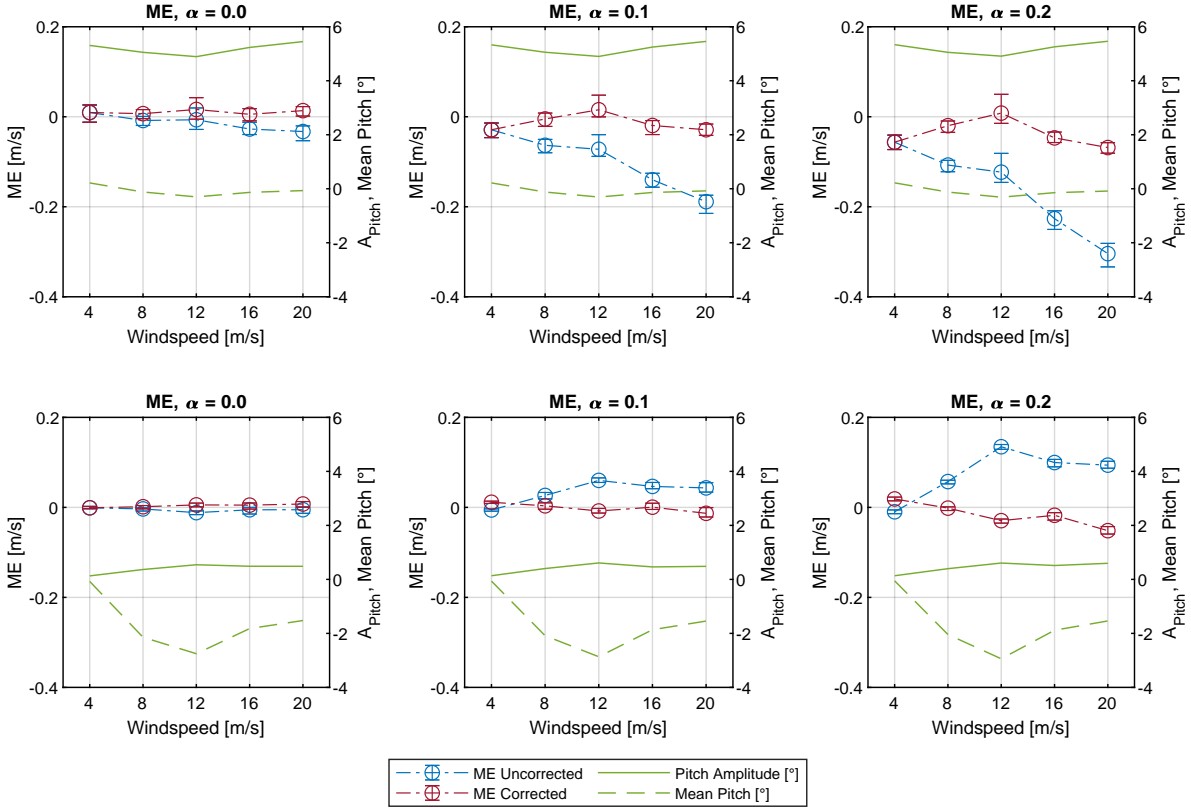

**Figure 13.** Mean errors of u- wind speed component for FLOATGEN (top) and WindCrete (bottom) FOWT with and without correction for varying vertical shear exponents.

## 5.2 Frequency filtering

We first test the filtering approach using the lidar simulation framework ViconDAR for random realizations of synthetic turbulent wind fields with prescribed floater dynamics.

Figure 14 shows an exemplary time series plot of lidar estimated u-component wind speed with and without application of the frequency filter. Additionally the u-component wind speed component of the original wind field averaged over the rotor plane is shown as a reference. Since the floater pitch frequency is defined by only one frequency in this case, a narrow peak in the resulting PSD of the lidar measured u-component is observed. Application of the bandstop filter corrects the pitch induced fluctuations accurately.

Figure 15 shows the MAE after application the frequency correction approach over wind speeds from 4ms$^{-1}$ to 20ms$^{-1}$ and different pitch frequencies for three different pitch amplitudes. As a reference, the figure shows the MAE without frequency





filtering. The remaining MAE after filtering is reduced to values of around 2% of the inflow wind speed. The variation of vertical shear exponent, pitch amplitude and floater mean pitch angle does not influence the MAE significantly.

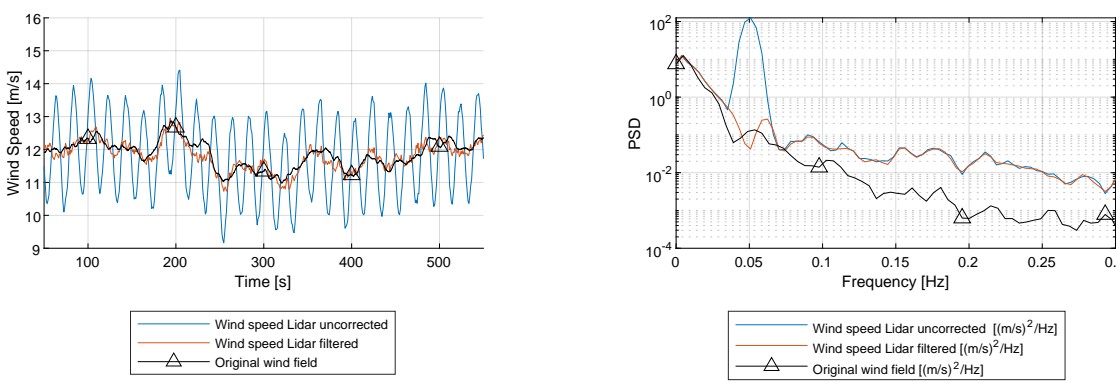

**Figure 14.** Left: Time series example of lidar estimated u-component wind speed, frequency corrected lidar u-component wind speed estimate and reference wind speed of the input wind field. Right: Power spectral density of lidar estimated u-component wind speed and frequency corrected lidar u-component wind speed estimate.

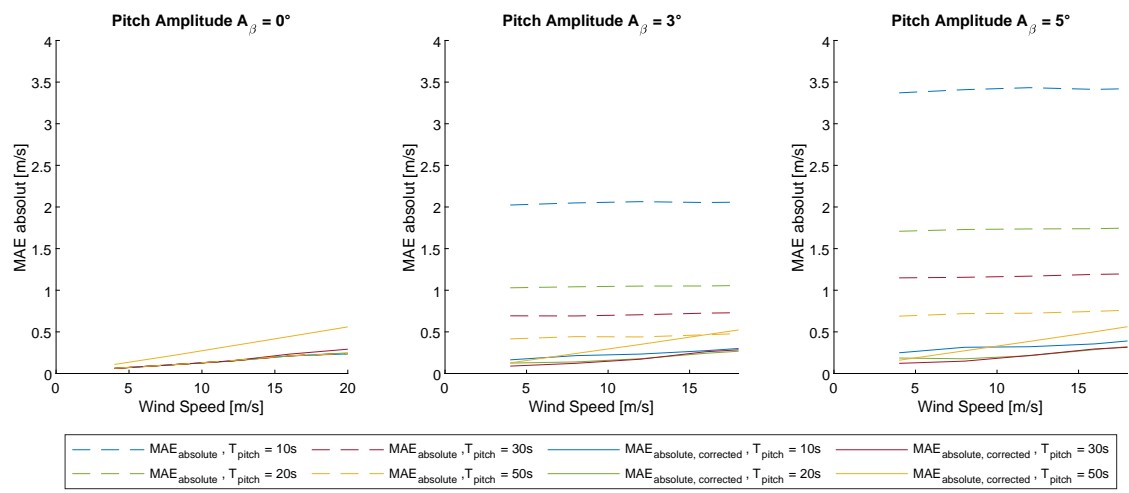

**Figure 15.** MAE of lidar u-component wind speed estimates with and without frequency correction for varying pitch amplitudes and vertical shear exponent of $\alpha = 0.1$. Fixed Parameters: $A_\psi = 0$ deg, $A_\gamma = 0$ deg, $A_{heave} = 0$ m.

In figure 16, example time series and corresponding PSD plots for simulated lidar measurements on the FLOATGEN and WindCrete FOWT before and after the application of the bandstop filter are shown. As a reference the wind speed average over the rotor plane is shown. The uncorrected measurements for FLOATGEN are characterized by periodic fluctuations





caused by the floaters response to wave excitation. Application of the frequency filter reduces the fluctuations significantly. For the WindCrete FOWT no clear influence of pitch motion can be observed in the time series and the corresponding PSD. Consequently, application of the frequency filtering cannot reduce the error between the lidar measured wind speed and the full wind field reference. The reference PSD of the original wind field represents an average over all points in the rotor plane. Therefore, the spectrum lies below the lidar estimated spectra.

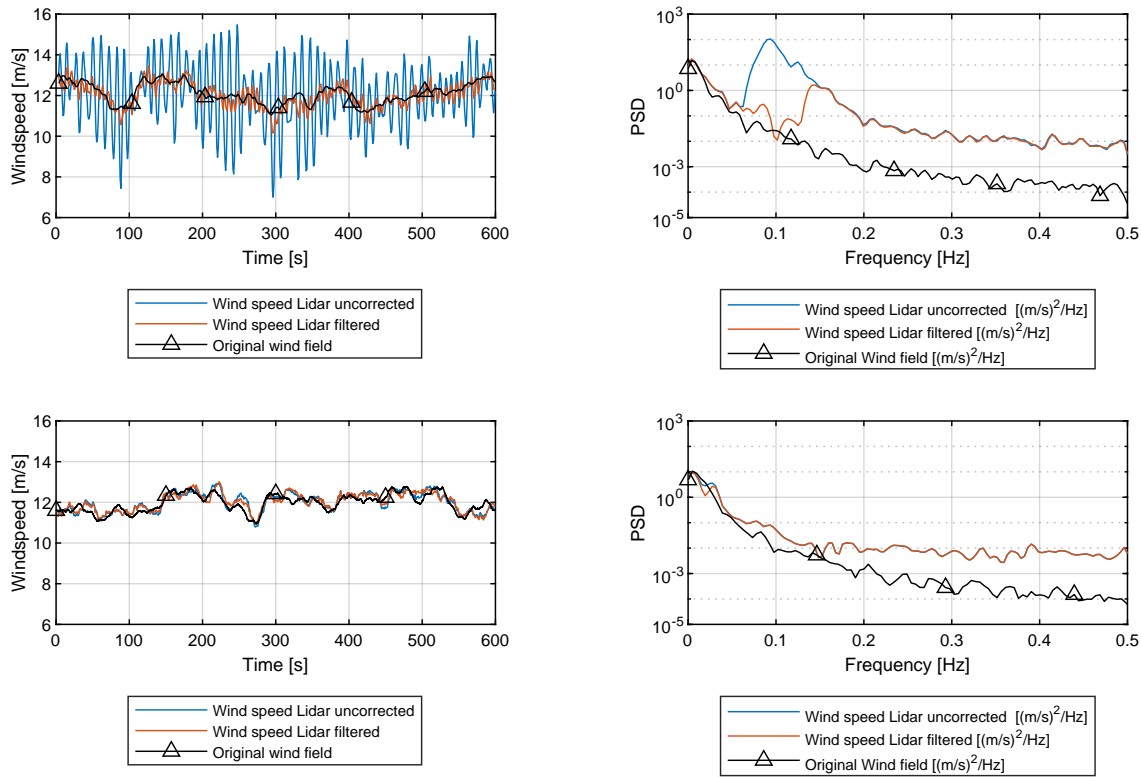

**Figure 16.** Time Series and PSD of simualated Lidar wind speed estimate with and without frequency correction. Top: FLOATGEN, Bottom: WindCrete.

Figure 17 shows the MAE results of the frequency filter based correction approach for simulated dynamics of FLOATGEN (first row) and WindCrete (second row) FOWT. For FLOATGEN the pitch motion of the floater is mainly determined by the wave conditions with high pitch excitation for wave frequencies, close to the natural pitch frequency of the floater. The frequency filtering of the present floater pitch frequency can significantly reduce the MAE.

     For the given wind and wave conditions the WindCrete Floater shows very small pitch excitation, resulting in small transla-
tional velocities of the nacelle and no significant peak in the frequency spectrum. In this case the MAE is not affected by the

 

frequency filtering approach. In contrary, the MAE of the filtered lidar wind speed estimates are slightly higher compared to the uncorrected value.

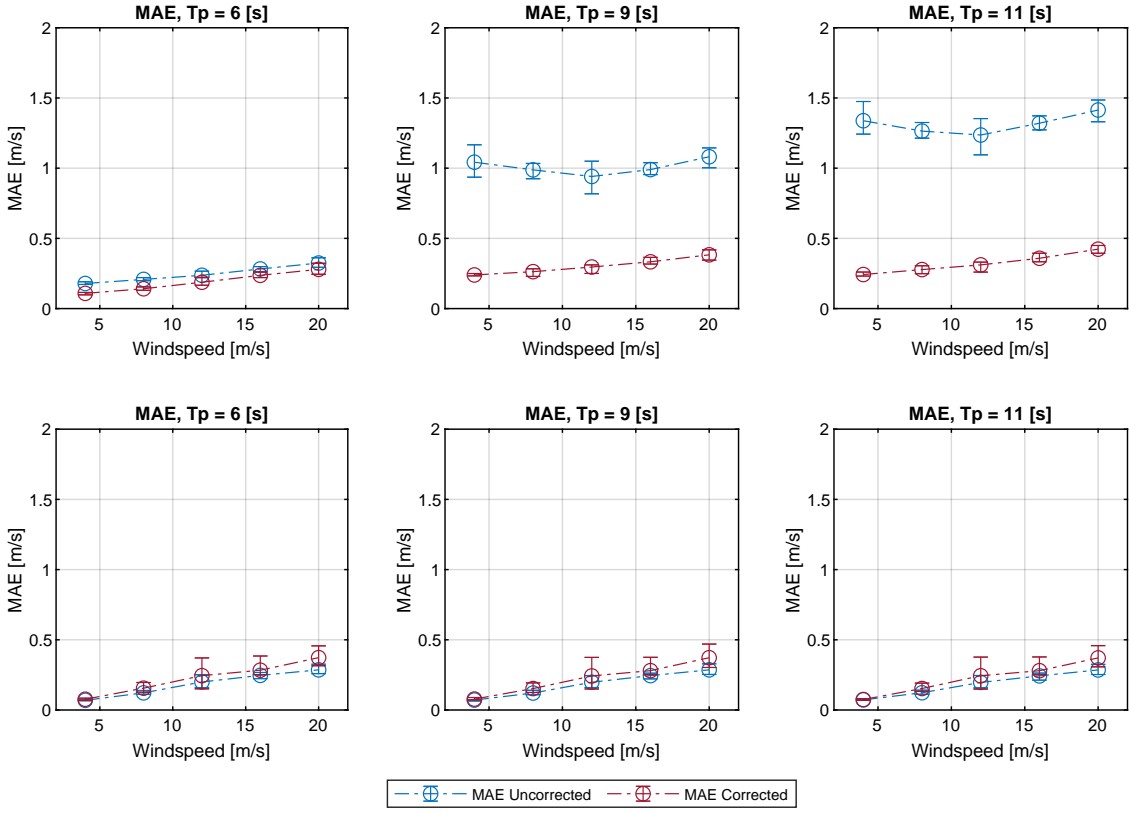

**Figure 17.** MAE of u- wind speed component for FLOATGEN (top) and WindCrete (bottom) FOWT with and without correction for varying wave periods and vertical shear exponent of $\alpha = 0.1$.

Besides MAE, the second metric used to evaluate the performance of the frequency filtering is ME. While for FLOATGEN, filtering of the pitch frequency does not introduce additional bias, for WindCrete significant bias of up to 0.2ms⁻¹ is introduced in the time series. A comparison of ME before and after application of the frequency filter is shown in figure 18. The MAE and ME results of this simulation study show, that the proposed frequency filtering approach for motion correction is able to reduce MAE while not introducing bias. This applies only to floaters that cause a distinct peak around the pitch frequency of the floater in the frequency spectrum of the lidar wind speed measurements.



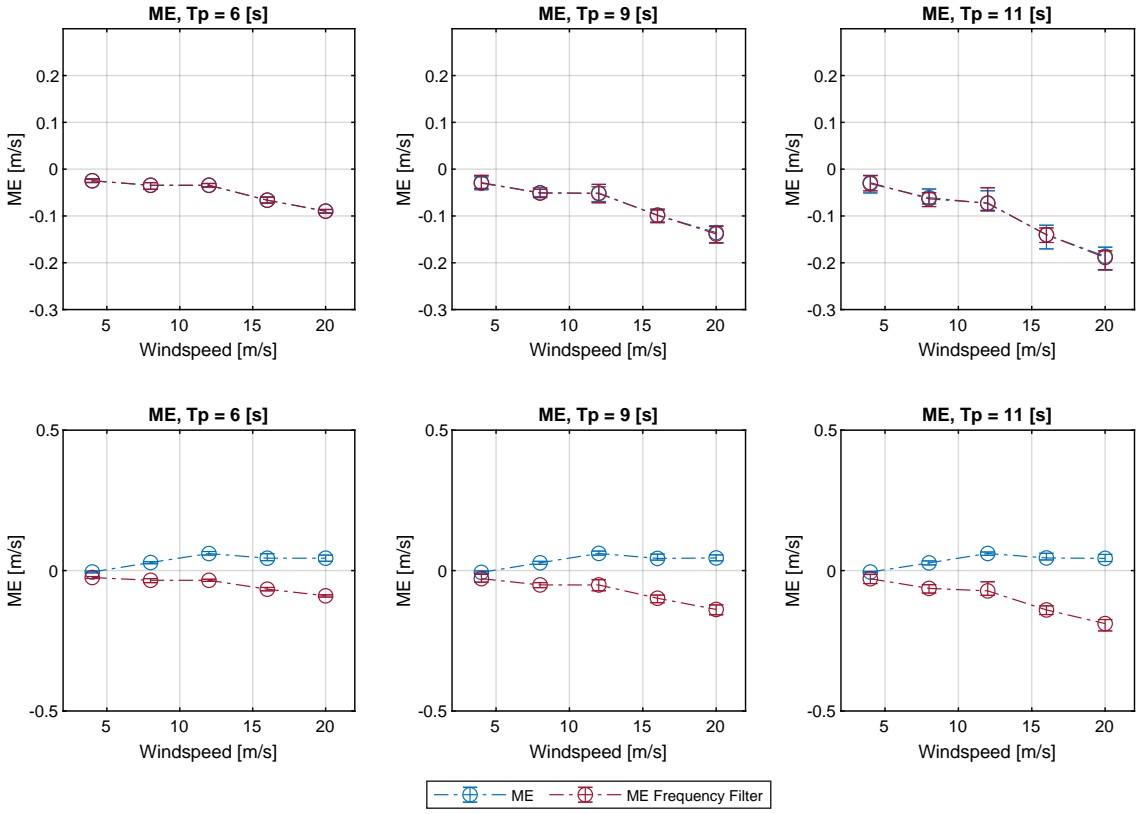

**Figure 18.** ME of u- wind speed component for FLOATGEN (top) and WindCrete (bottom) FOWT with and without application of frequency filter. Vertical shear exponent $\alpha = 0.1$.

## 6 Discussion

The motion influence on nacelle based lidar measurments was investigated with two different models. The introduced analytical model for the estimation of uncertainties and bias introduces several novelties and benefits compared to already existing lidar simulation and uncertainty quantification frameworks.

First, it specifically addresses nacelle based lidar systems on floating wind turbines for which very limited academic and industry experience exists and uncertainty quantification is crucial for the application in various use cases. The advantage of this
model over other, already available, simulation tools lies mainly in its simplicity. The assumption of a power law wind profile with no representation of turbulence does not require the generation of synthetic turbulent wind fields. The representation of floater dynamics in terms of frequency and amplitude parameters of the individual DOF avoids the need of numerical aero-elastic turbine-floater simulations. Thus, it allows an efficient estimation of motion induced uncertainties and biases based on basic design parameters of the FOWT concept and the lidar configuration. In this way computational expensive numerical



simulation can be avoided, while still considering most relevant effects of floater motion on the measurement. Based on these estimations and intended use of the lidar measurents, decisions about correction approaches can be facilitated. As shown in chapter 5.1 the use of model results for bias correction of measurements also shows additional potential for future use of the model.

The numerical model is much more sophisticated and considers several characteristics which are neglected by the analytical

model. In particular it uses synthetic turbulent wind fields to account for the turbulent nature of real wind fields. Additionally, the probe volume averaging effect of lidar measurements is considered. The lidar setup is represented in a more realistic way considering the temporal relation between the individual LOS measurements. However, comparing the results of the analytical and the numerical model we find good agreements between the two models which in general gives confidence in the quantification of uncertainties. Based on a parametric study it was found that the most influential floater DOF for nacelle based

lidar measurement is the pitch displacement leading to different effects relevant on different time scales. For time averaged measurements the pitch motion in combination with a vertical shear wind profile leads to a underestimation of wind speed. For FOWT configurations, operating at a non-zero mean pitch angle, shifted measurements positions in combination with a vertical shear profile lead to a overestimation of wind speed. Instantaneous wind speed measurements are mostly influenced by translational velocities of the nacelle, which are also caused by the rotational pitch movement of the floater. Two different

approaches for the correction of these effects were introduced and tested using the numerical lidar simulation framework. The model based bias correction approach is using bias estimates from the analytical model calculate correction values as function of pitch amplitude, frequency and present shear conditions.

For the testing case using prescribed dynamics based on amplitude and frequency parameters the correction approach yields very accurate results. Here, it should be noted that the analytical and the numerical model use the same dynamics inputs. In

reality the modelled floater dynamics of the analytical model might not exactly represent the real floater dynamics. Also the parameters determining the correction value for each 10 minutes are not constant for real floaters and need to be averaged, which adds uncertainties. However, the testing cases with aero-elastic simulation of a spar type and a barge type FOWT with very different pitch characteristics still yields good results with remaining mean errors of below 0.1 ms$^{-1}$. For all testing cases, it is assumed that the parameters necessary for the determination of correction values can be measured accurately. In reality

the determination of these parameters (mean pitch angle and pitch amplitude) would rely on inclination measurements which could add uncertainty. Thus, the application will need further verification with real measurements.

The correction of instantaneous fluctuations induced by fore-aft motions of the floater based on frequency filtering is tested for idealized conditions with prescribed dynamics as well as simulated dynamics of the two FOWT models. For idealized conditions and the assumed parameter space the frequency filter approach yields accurate results for pitch periods up to 30s.

For higher pitch periods, translational velocities of the nacelle caused by floater pitch motions are not increasing the MEA significantly. Thus the MEA is not predominantly determined by the pitch motion of the floater. Filtering of these frequencies yields increased MAEs, since relevant parts of the wind field spectrum are filtered out. Here, it should also be mentioned that the pitch frequency is assumed to be exactly known for each simulation run and not changing over time which does not accurately represent the behaviour of FOWTs under varying environmental conditions.





The testing cases using the simulated dynamics from FLOATGEN and WindCrete FOWT confirm the findings of the idealized conditions. For FLOATGEN, which has a relatively low natural pitch period of around 11 seconds and high pitch amplitudes the MEA is predominantly determined by the pitch motion of the floater. Here, the MAE can be reduced significantly using the filtering approach. For WindCrete with a natural pitch period of around 50s and smaller pitch amplitudes the filtering approach does not yield satisfactory results. For applications on real measurement data the filtering frequencies need to
be determined through measurements, e.g. based on a peak detection algorithm which might introduce additional uncertainties. Therefore, verification of the method with real measurements is necessary.

## 7  Conclusions

In this study we analyzed motion induced effects on lidar measurements from forward looking nacelle mounted lidars on FOWT. For this analysis we introduced a new analytical model for the estimation of uncertainty and bias in lidar estimated
wind speed. FOWT dynamics are modelled using amplitudes and periods of floater DOF in yaw, pitch, roll and heave direction. The deterministic wind field is modelled by a simple power law profile. Further we applied the GUM methodology to derive combined uncertainties in the LOS measurements as well as in reconstructed WFC. To verify the model outputs we compared the result to results uncertainties derived with the numerical lidar simulation framework ViConDAR. This lidar simulation follows a much more detailed modelling approach, in particular taking into account turbulent wind fields.

Results of a parametric study showed that the uncertainty of lidar estimated wind speed estimates are mainly caused by fore-aft motion of the lidar resulting from the pitch displacement of the floater. Therefore, the uncertainty is heavily dependent on the amplitude and the frequency of the pitch motion. On the other hand, the bias in 10 min averaged wind speed estimated is mainly influenced by the mean pitch angle of the floater and the pitch amplitude.

Further, we introduced two approaches for correction of motion induced effects. We used the analytical model to derive
a look up table of correction values for 10 min averaged wind speed measurements. Testing of the approach with simulated dynamics from two different FOWT concepts showed good results. Remaining mean errors between simulated lidar measurements and input wind fields were found to be below 0.1ms$^{-1}$ for both FOWT models. We used a frequency filter to correct fluctuations caused by floater pitch motions in instantaneous measurements. The correction can reduce the MAE in lidar wind speed estimates under certain conditions. The frequency filtering yields good results for dynamic conditions characterized by
harmonic pitch oscillation with low pitch periods and high pitch amplitudes. For dynamic conditions characterized by varying pitch oscillation or high pitch periods and low amplitudes, the frequency filtering cannot reduce MEA in lidar wind speed estimates.

*Code availability.* The analytical lidar uncertainty estimation tool is available on *"https://github.com/SWE-UniStuttgart/FLIDU"*. The numerical lidar simulation framework ViConDAR is available on *"https://github.com/SWE-UniStuttgart/ViConDAR"*

.





*Author contributions.* M.G. developed the analytical model for uncertainty estimation and the software implementation. M.G. conducted the simulation studies and drafted the paper. V.P. contributed with conceptualization, code review and review of the paper. J.G. and P.W.C. contributed with discussions and reviewed the paper.


*Competing interests.* The authors have the following competing interests: At least one of the (co-)authors is a member of the editorial board of Wind Energy Science.

*Acknowledgements.* This study has received funding from the European Union's Horizon 2020 research and innovation programme under the Marie Skłodowska Curie grant agreement N° 860879. Lidar measurement data was kindly provided by VAMOS Project, funded by the German Federal Ministry for Economic Affairs and Climate Action (BMWK) under the grant no. 03EE2004A.

## Appendix A

### A0.1 Analytical estimation of uncertainty in wind field characteristics

Following the GUM methodology for expression of uncertainty (Sommer and Siebert (2004)) the total uncertainty a output $y = f(x_1, x_2, ..., x_n)$ with $n$ uncorrelated input quantities $xi$ is given by:

$$U_y^2 = \sum_{i=1}^{4} \left(\frac{\delta f}{\delta x_i}^2\right) U_{x,i}^2 \tag{A1}$$

Where $U_{x,i}$ is the standard uncertainty of input quantity $x_i$.

Accordingly the total uncertainty in the LOS measurements, induced by floater dynamics is given by:

$$U_{v_{los}} = \sqrt{\left(\frac{\delta v_{los}}{\delta \psi}\right)^2 U_\psi^2 + \left(\frac{\delta v_{los}}{\delta \beta}\right)^2 U_\beta^2 + \left(\frac{\delta v_{los}}{\delta \gamma}\right)^2 U_\gamma^2 + \left(\frac{\delta v_{los}}{\delta h_{heave}}\right)^2 U_{h_{heave}}^2 + \frac{\delta v_{los}}{\delta \boldsymbol{x_{vel}}}^2 U_{\boldsymbol{x_{vel}}}^2} \tag{A2}$$

Where $U_\psi, U_\beta, U_\gamma, U_{h_{heave}}, U_{x_{vel}}$ are the standard uncertainties of input quantities. The formulation in equation A2 assumes that the considered input quantities are uncorrelated. As the input quantities represent the dynamics of the floater which are closely connected to wave forces acting on the floater, they are modelled as harmonic oscillations. Following Sommer and 620 Siebert (2004) the standard uncertainties of this type of input quantities are given by:

$$U_i = \frac{\Delta a}{\sqrt{2}} \tag{A3}$$

where $a_i$ is the amplitude of the assumed oscillation of the input quantity $i$. The derivatives in equation A2 are the partial derivatives of equation 10 with respect to the considered input quantities. The solutions of the partial derivatives can be found in appendix A0.4.



## A0.2 Uncertainty propagation through wind field reconstruction

In this section we derive uncertainties in reconstructed wind field characteristics based on previously derived LOS uncertainties. Since the lidar is only able to measure the wind speed in the line of sight direction, wind field characteristics including the horizontal wind speed components need to be reconstructed from the LOS measurements. It is not possible to reconstruct the 3-dimensional wind vector from a single LOS measurement unambiguously. Different approaches to this problem like the Velocity-Azimuth Display technique introduced by Browning and Wexler (1968) are discussed in literature. Other approaches combine several LOS measurements and use assumptions about spatial and temporal correlations between the measurements to reconstruct the wind field characteristics (see e.g.Borraccino et al. (2017)).

For the analytical derivation of uncertainty we employ a simple wind field reconstruction algorithm which assumes the $v$ and $w$ component of the wind field to be zero. With this assumption the reconstructed u-component of the wind speed at each beam is given by:

$$u_i = \frac{V_{LOS,i}}{x_{L,i}r_i} \tag{A4}$$

with $V_{los,i}$ being the LOS wind speed of beam 1 to 4, $x_{L,i}$ being the focus point x- coordinate of beam $i$, $r_i$ being the measurement distance. Further we combine the measurements of all four beams by averaging the wind speed estimates of the individual beams.

Thus, the reconstructed wind speed component $u_{rec}$ is given by:

$$u_{rec} = \frac{1}{4} * \left( \frac{V_{LOS,1}}{x_{L,1}r_1} + \frac{V_{LOS,2}}{x_{L,2}r_2} + \frac{V_{LOS,3}}{x_{L,3}r_3} + \frac{V_{LOS,4}}{x_{L,4}r_4} \right) \tag{A5}$$

Since the reconstruction approach takes into account LOS measurement of all four beams for one estimate of $u_{rec}$, these estimates do not represent the wind speed at one focus point, but an average over the measurement plane.

Again, the uncertainty of the reconstructed horizontal wind speed components can be estimated combining the standard of uncertainties of the input quantities by following the GUM methodology. The combined uncertainty of $n$ correlated input quantities is given by:

$$U_y^2 = \sum_{i=1}^{4} \frac{\delta f}{\delta x_i}^2 * U_i^2 + 2 \sum_{i=1}^{N-1} \sum_{j=i+1}^{N} \frac{\delta f}{\delta x_i} \frac{\delta f}{\delta x_j} U_i U_j r_{i,j} \tag{A6}$$

Where $r_{i,j}$ is the correlation coefficient of beam $i$ and $j$. Considering equation A5 the reconstructed wind speed component $u_{rec}$ a function of the four LOS velocities. Thus the total uncertainty of $u_{rec}$ given by:

$$U_{u_{rec}}^2 = \sum_{i=1}^{4} \frac{\delta u}{\delta v_{los,i}}^2 * U_{los,i}^2 + 2 \sum_{i=1}^{N-1} \sum_{j=i+1}^{N} \frac{\delta u}{\delta v_{los,i}} \frac{\delta u}{\delta v_{los,j}} U_{los,i} U_{los,j} r_{i,j} \tag{A7}$$

and



Where $U_{los,i}$ is the standard uncertainty of beam $i$ as calculated in equation A2. In this case the LOS velocities of the four beams cannot be assumed uncorrelated. The correlation is considered by the correlation coefficient $r_{i,j}$ of the LOS velocities of beam $i$ and $j$. The partial derivatives can be found in appednix A0.4.

The correlation coefficients between the LOS measurements of the individual beams are needed as an parameter for calculation of uncertainty of reconstructed wind field characteristics. They are influenced by the changing LOS directions and the position of focus points in space and the assumed wind field. Thus, correlation coefficients are dependent on the set of dynamic input parameters as well as the phasing between the single DOF. In the model, the correlation coefficients are calculated for the present set of model input parameters. This is done by evaluating equation 10 over time for each LOS and calculating the
correlation between the resulting LOS time series.

### A0.3    Analytic Estimation of Bias

The lidar measurement model introduced 2.2 is also used to estimate systematic biases in reconstructed wind field characteristics which occure due to floater dynamics. Using equation 10, which combines the temporal evolution of all dynamic input quantities the LOS velocity is calculated for a defined parameter space over a defined time span. Further, the wind field re-
construction approach introduced before is applied to calculate the horizontal wind speed component $u_{rec}$. The Bias of the reconstructed u- component wind speed is calculated by

$$Bias_u = u_{ref} - u_{rec,mean} \tag{A8}$$

where $u_{ref}$ is the reference wind speed and, $u_{rec,mean}$ is the mean value of the reconstructed wind speed component. The reference wind speed is calculated as average over a rotor plane. The rotor diameter for this hypothetical rotor is chosen to
be $d_{rotor} = 150m$. The results of the bias estimation are very sensitive to the exact calculation procedure of the reference value. Therefore, to make the results comparable to the numerical model results, the calculation procedure is the same as in the numerical model. In the numerical model the u- wind speed components of all grid points of the synthetic wind field which are within the rotor plane is averaged. For the the reference values $u_{ref}$ of the analytical model the wind speed components of the same grid points are calculated according to the assumed power law wind profile. The average of these points is taken to
calculate $u_{ref}$.

     Equation 10 combines the temporal evolution of all dynamic input quantities. Therefore, different effects can potentially influence the the estimation of bias. First it must be ensured that the length of the time window of calculation is large enough to avoid any relevant influence of time window length on the bias estimation. Second the phasing between the individual dynamic input quantities could influence the bias estimation.

Figure A1a) shows the standard deviation within 100 realizations of the bias estimation of the wind speed u-component for one set of input parameters as a function of time window length. The phasing of input dynamics (yaw, pitch, roll and heave) is randomized for each realization. It can be seen that the standard deviation is fluctuating over the time window length depending on the ratio between time window length and dynamics frequencies with minimums for time window lengths being multiples of dynamics frequencies. This effect is small for longer time window lengths where the averaging period is one order of





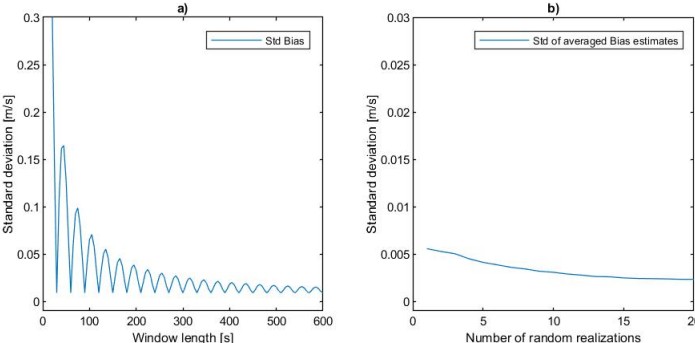

**Figure A1.** a) Standard deviation within of bias estimation within 100 random realization over time window length. b) standard deviation of averaged bias estimations over number of averaged random realizations, time window length 600s.

magnitude higher than the period of input dynamics. The remaining fluctuation in standard deviation for a time window length of 600s is small and can be neglected. However, the standard deviation is not converging to zero, since the random phasing between the individual dynamics input quantities is part of the standard deviation. Therefore, several random realizations with a time window length of 600s are averaged. Figure A1b) shows the standard deviation of averaged bias estimates as function of the number of random realizations, averaged for one bias estimate. For the remainder of the work all bias estimates of the 690    analytical model are calculated for a time window length of 600s and averaged over 10 random realizations.

### A0.4    Partial derivatives

$$\frac{\delta v_{los}}{\delta \psi} = -V_{ref}\left(\frac{z_L + h_{lidar} + h_{heave}}{H_{ref}}\right)^{\alpha}(\sin\varphi(x_L\sin\psi + y_L\cos\psi)) + \cos\psi(y_L\sin\psi - x_l\cos\psi) \tag{A9}$$

$$\frac{\delta v_{los}}{\delta \beta} = \frac{\alpha V_{ref}(-x_L\cos\beta - z_L\sin\beta)(sin\varphi(x_L\cos\beta + z_L\sin\beta) + y_L\cos\varphi)\left(\frac{h_{lidar}+h_{heave}-x_L\sin\beta+z_Lcos\beta}{H_ref}\right)^{\alpha-1}}{H_{ref}}$$
$$+ V_{ref}\sin\varphi(z_L\cos\beta - x_L\sin\beta)\left(\frac{h_{lidar} + h_{heave} - x_L\sin\beta + z_Lcos\beta}{H_{ref}}\right)^{\alpha} \tag{A10}$$

$$\frac{\delta v_{los}}{\delta \gamma} = \frac{\alpha V_{ref}(y_L\cos\gamma - z_L\sin\gamma)(x_L\sin\varphi + \cos\varphi(y_L\cos\gamma - z_L\sin\gamma)\left(\frac{h_{lidar}+h_{heave}+x_L\sin\gamma+z_Lcos\gamma}{H_{ref}}\right)^{\alpha-1}}{H_{ref}}$$
$$+ V_{ref}\cos\varphi(-y_L\sin\gamma - z_L\cos\gamma)\left(\frac{h_{lidar} + h_{heave} + y_L\sin\gamma + z_L\cos\gamma}{H_{ref}}\right)^{\alpha} \tag{A11}$$

$$\frac{\delta v_{los}}{\delta h_{heave}} = \frac{\alpha V_{ref}\left(\frac{h_{heave}+h_{lidar}+z_L}{H_{ref}}\right)(x_L\sin\varphi + y_L\cos\varphi)}{h_{heave}+h_{lidar} + z_L} \tag{A12}$$





$$\frac{\delta v_{los}}{\delta x_{vel}} = x_I \qquad \frac{\delta v_{los}}{\delta x_{vel}} = y_I \qquad \frac{\delta v_{los}}{\delta z_{vel}} = z_I \tag{A13}$$

$$\frac{\delta u}{\delta v_{los,i}} = \frac{1}{4}\frac{1}{x_{L,i}} \tag{A14}$$





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
