# Peer review of "Quantification and correction of motion influence for nacelle-based lidar systems on floating wind turbines"

_Wind Energy Science, 2023_

## Author Comment (AC1)

**Authors' Response Preprint wes 2023-11**

We thank all reviewers for their constructive comments and suggestions which helped greatly in improving the manuscript. We highly appreciate the time and effort the reviewers dedicated to this. In the following, we reply to the reviewer's comments point by point. Original comments are given in black, and answers in blue.

On behalf of the authors

Moritz Gräfe

**Reply to RC-1**

The authors would like to thank Felix Kelberlau for his constructive and useful feedback. All comments have been taken into consideration for the revised version of the manuscript. A list of individual comments and replies follows:

**General Comments**

The title of the manuscript clearly reflects the content of the work with regard to motion correction, but it understates the significant amount of work done towards quantification of the measurement error.

We agree with this comment and have changed the title to:

"Quantification and correction of motion influence for nacelle based lidar systems on floating wind turbines"

The objectives and methods of the study are clearly outlined in the introduction and described in enough detail to make the study reproducible. All assumptions that are made for modelling the lidar measurements are stated explicitly. Though, several of the assumptions made are not or not sufficiently reasoned. They are also not well discussed in terms of their implications for the results. This should be improved in a revised version of the paper.

As detailed in the specific comments, assumptions made in the manuscript are justified and discussed in more detail in the revised version.

Many simulation results are presented and briefly discussed. The paper would benefit from a more concise presentation of the most important results and a more in-depth discussion that focuses on the results that are of practical relevance.

We agree with this general comment and have shortened the results section of the manuscript. Figures 11, 12, 15 and the associated texts have been removed. Instead, the discussion of the remaining figures has been revised.

Overall, the presentation of the work has a clear structure and is well written. The overall good readability is interrupted by very many minor mistakes regarding missing dashes, commas, and prepositions.

An editorial revision has been carried out.

**Specific Comments**

**Introduction**

In general, relevant literature has been cited and described. The authors should also include the main findings and limitations of the cited references, so that existing knowledge gaps become clear that motivate the paper. Please explain explicitly why your study extends the existing body of literature.

The introduction of the manuscript has been revised, with a special focus on existing research gaps and the contribution of the manuscript.

l 11: The abstract should be improved by making it more concise and result oriented. For example "Further, we discuss the need for motion compensation..." should be replaced by for example "We find that motion compensation is needed if...".

Following the reviewer's suggestion, the abstract has been revised and shortened.

l 71-75: This analysis of an error cause should better be moved to section 3 where it can be put into context that is still missing in the introduction.

After consideration of the suggestion, we decided to keep the section at its original position in the manuscript. It represents findings from another study, and we believe it could be confusing to introduce it during the discussions of findings from this study.

l 91: Better include the objectives to be the conclusion of your literature research instead of a short separate subsection.

Following the reviewer's suggestion, the objectives section has been included at the end of the introduction. (l 101)

l 95: A third objective is probably the assessment of the introduced correction methods.

We agree with this statement. A third objective, explicitly stating the point mentioned above has been included. (l 104)

**Methodology**

The methodology separated into analytical and numerical modelling is well described. Subsection 2.1 is missing information about time and duration of the campaign, also relevant information about environmental conditions during the campaign are missing (calm or rough sea states...). Some of the notation introduced in 2.2 and 2.3 is incoherent or unclear. Please revise.

Details about the measurement campaign have been added in section 2.1 (l 133). Unfortunately, due to confidentiality, details about the sea-state cannot be given. Since the measurement data is only used as an example to show that there is motion influence in the lidar measurements, we believe this is acceptable.

The notation in section 2.2 and 2.3 has been revised and made consistent. Additionally, the order of the derivation of equations has been changed for better readability.

l 129: Give a reference to where in the manuscript the lidar is fully described (beam timing, range gates...)

Done

l 145f: This is probably a reasonable simplification but the authors must explain why surge and sway can and should be omitted. Are they small compared to the rigid body motion due to tilt of the floater?

The authors agree, that this simplification should be justified. The following explanation has been added (l 156):

"The low-frequency displacements of the floater in surge and sway direction are typically causing slow translational velocities of the nacelle compared to the effect of rotational displacements of the floater. Therefore, the sway and surge displacement of the floater are not considered in the model as individual DoF."

The authors would also like to point out, that translational displacement in surge and sway direction would have no influence on the measurement due to the assumption of a vertically homogenous wind field. Translational velocities which could influence the measurement could be added easily to the vector given in equation 3. The partial derivatives of VLOS with respect to the translational velocities would remain unchanged.

l 148: Variable "a" must be introduced.

Done

l 155: Refer to Figure 2 and introduce x_trans and z_heave

Done

l 165: Please describe: What are the implications of these simplifications on the measurements? Why are they acceptable?

The following sentence has been added as an explanation (l 182):

"While these assumptions avoid the introduction of a turbulence model and enable the analytical derivation of uncertainty following the GUM methodology, it should be pointed out that model results do not reflect any effects originating from the turbulent nature of real wind fields."

Additionally, the model results have been compared to the numerical modelling approach, where no significant deviations have been found.

l 167: The "u-component" of the wind vector is often referred to in the manuscript but it is never introduced. The authors should define it in the methods section.

Done (l 197)

l 172: Mention the specific beam geometry here or refer to where in the manuscript it is given.
Done

l 175: Please include the rotation matrix Rx,y,z (or R(ψ,β,γ)), instead of just mentioning it.
The rotation matrix has been added to the appendix and a reference is included (l 170).

l 180: Briefly describe why volume averaging is omitted (or refer to 2.3 where it is mentioned). Also why is turbulence not needed in the analytical model?

Probe volume averaging effects are not considered in order to simplify the analytical uncertainty derivation and avoid overly complicated expressions in the partial derivatives. It was expected that probe volume averaging effects are not significantly influencing uncertainties induced by floater motions. This is confirmed by the comparison of analytical and numerical results.

The sentence (l 199) has been changed to :

"To enable a straightforward analytical uncertainty derivation probe volume averaging effects are not considered."

An explanation concerning turbulence has been added in the answer to the comment for (l 165)

l 208: "surge", "sway" and "heave" have been introduced and should be used for consistency with the rotational DoF.

Done

Eq. 11: What is the index I referring to?

The notation, in particular the indexing of coordinates, has been revised throughout the manuscript.

Eq. 12: Index "P" or "p" (Figure 3)

The notation, in particular the indexing of coordinates, has been revised.

Fig. 3: This figure needs improvements: By definition [xI,yI,zI] are the current positions of the lidar focus points. And [xp,yp,zp] are the positions of focus points after translation. What is the dashed blue vector? If it shall be the translation vector, why does it not connect [xI, yI, zI] and [xp,yp,zp]? Things stay a bit unclear to me.

The figure and the notation of focus points and coordinate systems have been updated for clarity. [x_pI. y_pI, z_PI] denotes the position of focus points in the I-coordinate system. The dashed blue vector is intended to show the translation between the I- and L- coordinate systems.

l 220: Stick to the x,y,z order: "surge, sway and heave".

Done

**3 Motion influence in nacelle based lidar measurements**

Both 3.3 and 3.4 would benefit from a concise summary of findings like it is given at the end of 3.2. Which parameters are the most critical for uncertainty and bias? What are the ranges of error caused by the different analysed parameters?

3.3: The following summary has been added at the end of the subsection (l 349):

"This analysis shows that the uncertainty in reconstructed u-component wind speed measurements is dominantly determined by the fore-aft motion of the nacelle, which is mainly caused by the pitch rotation of the floater. High pitch amplitudes and high frequencies cause high uncertainties in wind speed estimates. For the conditions and models considered this uncertainty is found to be in the order of magnitude of 1 m/s."

3.4: The following summary has been added at the end of the subsection (l 382):

This analysis shows that the bias in reconstructed u-component wind speed is determined by the mean pitch angle of the floater and the pitch amplitude. In the presence of vertical wind shear, negative mean pitch angles cause upwards shifted focus points which result in positive bias in u-component wind speed estimates. High pitch amplitudes cause negative bias in u-component wind speed estimates. This bias is in the order of magnitude of 0.1 m/s.

l 230: Use subplot labels, e.g., "(a)" to refer to parts of a figure. This accounts for several instances througout the manuscript (e.g., l 341).

The suggestion has been implemented in several figures throughout the manuscript.

l 238: For the frequently occurring wind speed range around 10 m/s, the pitch standard deviation (approx. 0.5deg) is lower than the mean pitch angle (approx. 0.6deg). From this, it is not intuitive that there will be "significant fluctuations in the lidar measurements" while "no large errors in the mean wind speed estimates due to the mean pitch angles are expected". Please explain why you assume stronger influence from the dynamic motion. The authors can do so for example by adding to the plot the translational velocity of the lidar caused by tilt.

We agree with the reviewer that this is not immediately intuitive. The standard deviation of the translational velocity in x-direction, estimated from the pitch time series signal, has been included in the plot. The related text has been revised (l 256).

Fig. 4 caption: Left: Describe error bars in left plot. What do they show? Right: Which LOS velocity? (Is it arbitrary?). In general, give more information in the captions (here: Data from measurements or simulation?), so that readers who first don't read the text get a better impression.

The caption has been corrected. All captions have been revised, as suggested by the reviewer.

l 246: Second peak is not visible in the spectrum which ends at 0.5 Hz. Instead, second peak is visible at 0.33 Hz?

Accidentally, the text in the first manuscript referred to an old version of the figure, which showed a peak at the tower bending frequency in the pitch signal. Since we want to focus on the most important effect of motion on the lidar measurements, we decided to show the frequency plot only up to half the lidar sampling frequency (0.5Hz). The text concerning the tower bending frequency has been removed.

l 253: How are uncertainty and bias defined? Maybe refer to the definitions in the appendix.

A reference to the Appendix has been included.

l 263: The authors should mention that the first analysis presented here is valid only for steady displacement (or very slowly fluctuating motion << 1Hz)

The following explanation has been added (l 286):

"In this analysis, quasi-static displacements are assumed. Velocity components resulting from changing quasi-static conditions are not considered."

l 263: "of [displacement] in individual DOF"

Done

Fig. 5: Use differentiating text for the subplots in one column (e.g., "Yaw, α=0.0", "Yaw, α=0.1"...). All plots with zero motion but shear show mean reconstructed u velocities < reference wind speed. I understand that this is due to averaging over the rotor plane but it is not intuitive and makes interpretation of the entire figure difficult. Consider normalizing the y-axis to wind speed at zero motion. Or at least add the recontructed wind speed at zero motion to the plots as horizontal lines.

The reviewer's suggestions have been implemented. The y-axis of the plots is now normalized to the zero-motion reconstructed wind speed for each subplot. Titles and captions changed accordingly.

l 270: "stays nearly constant" (roll leads to slightly reduced vertical measurement positions)

Agreed and implemented.

l 275: This could be better described with the help of the chosen vertical shear model (Eq. 4): The effect of heave leads to identical sinusoidal variations of the measurement elevation for all four beams. As a result the reconstructed wind speeds in u-direction will show a minor negative bias, i.e, lidar-measured wind speeds will be lower than reference wind speed. This is because the horizontal wind speeds increase slower with increasing height than they decrease with decreasing height. The authors could also give an example to prove the insignificance of this bias in comparison to other biases. Without such a prove the reader might not believe that "no significant bias is introduced by the heave displacement".

We agree that the effect of heave elevation theoretically leads to a negative bias. The explanation has been changed to (l 294):

"Heave displacement in combination with non-linear vertical wind shear profile causes fluctuation in the reconstructed wind speed due to changing measurement elevation. As a result, the reconstructed u-component wind speed will show a small negative bias because horizontal wind speeds increase slower with increasing height than they decrease with decreasing height. However, for the expected range of floater heave elevations, this effect is small compared to the effect of floater pitch motion."

l 277 f: It must be clarified that this is true for slow motion relative to the 1Hz scanning pattern of the lidar.

The plot intends to show the influence of translational velocities in quasi-static conditions. No change in translational velocity is considered. The following explanation has been added to the manuscript (l 299).

"Figure 6 illustrates the dependency of measured LOS velocities and reconstructed wind speed on translational velocities of the lidar device for quasi-static conditions. No fluctuation of translational velocities is considered."

Fig. 6 caption: Describe the value of α.

Equivalent to Fig. 5, the y-axes have been normalized, and shear values are given in the subplot titles as well as in the caption.

l 301: Figure 7 shows uncertainties "up to 15%", not "up to 20%"

corrected (l 323)

Fig. 8: Use a),b),c), and d) as titles above the subplots and refer to Table 2 in the caption. Otherwise, it is difficult to get all relevant information and interpret the Figure.

Subplot titles have been added as suggested and a reference to table 2 is included. Shear variation has been added to table 2.

l 324: Isn't it straightforward to show that this relationship is reciprocal: Half the period-> double the translational velocity -> double the uncertainty? The authors can consider using Pitch frequency instead of Pitch period for the horizontal axes in Fig. 8 to show a linear relationship.

We agree that the relationship is reciprocal. Since several other figures in the paper use periods instead of frequencies, we decided to keep using the pitch period in figure 8 for consistency.

l 354: This is 0.5% of the measurement value and would be very significant. In this case the deviations between analytical and numerical solution would need to be explained. From the Figure, it looks like the deviations are actually below 0.02m/s.

Indeed deviations are in the range of up to 0.015m/s or 0.1%. This has been corrected (l 381)

**4 Correction Approaches**

l 358-364: Good to summarize but as an improvement, consider to describe the three effects in terms of uncertainty and bias separately. Maybe use a table with DOFs (vertical) and

uncertainty and bias (horizontal). Then fill the cells with strong (++,--), low (+,-) and no effect (0).

We agree that the main findings should be discussed separately. However, we believe that the introduction of a new quantification (++,+,0) system could be inaccurate. We decided to keep the structure with the bullet points and divide them into uncertainty and bias parts as suggested by the reviewer (l 391).

l 370: "nacelle-based lidar"

The notation "nacelle-based" has been implemented throughout the paper.

l 396: How small are "small time scales"?

The sentence (l 429) has been changed to:

"The application relies on instantaneous time series information of the inflow wind speed with sampling frequencies in the region of 1Hz."

l 408: The authors should explain their motivation for not "correcting lidar measurement time series based on the instantaneous turbine tilt angles". Although this would offer a chance correcting for bias and uncertainty at once. What is advantageous about the look-up table plus frequency filtering? What are the drawbacks?

The following discussion has been added in the section "Model based correction" (l456)

"The approach avoids the need for synchronized motion time series data, which might not be available in all cases. Inclination sensor signals might be noisy or not accurate enough due to the influence of nacelle acceleration on the sensor. Thus, the suggested approach is easy to implement for practical applications. However, it relies on the availability and accuracy of floater dynamics statistics Inaccuracies could occur in transient conditions, where mean floater dynamics, do not represent the actual floater dynamics sufficiently."

Eq. 13, l 413: "v_correction" is later called "v_corr".

Done

Table 7 caption: Introduce "Hs" and "Tp" here.

Done

**5 Results**

l 465-475: Consider removing these three short paragraphs including Figs. 11 and 12. Another parameter study does not add new knowledge and the efficacy of the model-based correction will be shown in Fig. 15. If the authors decide to keep the figures, they should be combined into one

We agree, that the added information from these figures is limited. In order to shorten the manuscript overall, figures 11, 12, 15 and the associated text have been removed.

l 482 & 489: Give ME in % here. Relative error is less dependent on WS.

Percentages for the ME values have been added to the text (l 500, l 507).

Fig. 13:  Adjust both y axes to use the same grid lines: For example:

y left: [-0.4 0.2]

y right: [-6 6]

The suggestion has been implemented.

I recommend showing the relative error.

Since throughout the manuscript absolute error values are used for quantification, the authors decided to keep the figure with absolute error values for consistency. As suggested, relative error values have been added to the text.

l 498ff: Consider deleting this paragraph and Fig. 15. It does not contain new ideas or knowledge.

We agree with the statement and removed Fig. 15 and the associated text.

l 505: I agree. Please add: How do the authors assess the remaining motion-induced error? (Why) Is it acceptable?

Mean Errors after the application of the frequency filter are shown in Figure 15 (revised manuscript). The plot shows that the application of frequency filtering does not introduce additional mean error for the case of the FLOATGEN concept, but does introduce mean errors for the WindCrete Concept. In our study, it is not evaluated, if the remaining MEA after application of the frequency filter is acceptable in general, as this would depend on the intended use of the data. Therefore, to avoid losing the focus of the manuscript we decided not to include this discussion.

l 509: Put this into a wider context also considering lidar-specific effects: line-of-sight averaging and cross contamination caused by combining spatially-separated measurement volumes.

The lidar spectrum does contain the effect of probe volume averaging, and cross-contamination. (as far as it is represented in the simulation)

Please see the figure below for your reference.

[Figure]

Here, the violet line represents the PSD of the four grid points, corresponding to the lidar focus points. It can be seen, that this line lies above the lidar spectrum. This effect can be attributed to the effect of probe volume averaging in the lidar measurements.

The following sentence has been added to the manuscript (l 523).

"It should be noted, that the lidar spectrum does contain the combined effect of probe volume averaging and cross contamination due to spatially separated measurement volumes."

Fig. 16 caption: Is wave case 1,2 or 3 shown?

Wave case 3, added to the caption

l 512: Which is the natural pitch frequency? Please add the value, if known.

Done

l 516: Why is the MAE of the filtered lidar wind speed estimates slightly higher than the uncorrected value? Also, how do you explain that the remaining error after correction is approximately the same for all cases?

The following explanation has been added (l 531):

"On the contrary, the MAE of the filtered lidar wind speed estimates is slightly higher compared to the uncorrected value, because the applied frequency acts on the measured wind field spectrum itself. Thus, not only motion-induced frequency components are filtered. MAE values are not significantly influenced by wave conditions because of the small dependency between floater pitch response and different wave conditions."

**6 Discussion**

l 541: The authors should discuss what is the effect of including turbulence, volume averaging and the real scanning pattern.

The following explanation has been added (l 559):

"A comparison of the results from the analytical and numerical model shows good agreement between both models. This indicates that the combined effect of turbulence, probe volume averaging and time relation between the LOS measurements is small. The individual influence of these characteristics cannot be derived from our results, since no sensitivity study was conducted."

l 561: I agree. Many practical issues like time-synchronization and measurement accuracy are not considered here. Please explain the possible consequences for the application of your method.

We agree, that different practical issues are not considered in the theoretical study. However, we believe that the proposed approach is relatively robust compared to methods that require synchronized motion time series measurements. In our case, only the mean amplitude and mean pitch angles are needed. These quantities could also be predefined as a function of environmental and operating conditions (wind speed, sea-state, turbulence, shear). Since no sensitivity study on the variation of input parameters was performed, we cannot quantify the effect of errors in the estimation of input parameters. We acknowledge that the investigation of the issue would be an interesting topic for a follow-up study.

The following explanation has been added to the manuscript (l 579):

"In reality, the determination of these parameters (mean pitch angle and pitch amplitude) would rely on inclination measurements which could add uncertainty to the corrected wind speed estimates. A sensitivity study, quantifying the effect of these uncertainties and further verification with real measurements is needed before application of the methodology."

**7 Conclusion**

l 588: In my opinion the wind shear coefficient should be added to the list of parameters determining the bias.

We agree with the statement and added shear to the list of parameters (l 606)

**Appendix**

l 618: At least pitch deflection and velocity in x direction are correlated. Who could this influence the results?

We agree that the two mentioned quantities have a strong dependency. The velocity in the x-direction is determined by the time derivative of the pitch deflection angle. However, mathematically, the correlation coefficient between pitch deflection and x-velocity caused by the pitch deflection is zero due to a phase shift of 90 degrees. X-velocity is zero at peak pitch

deflection and vice versa. Therefore, in this case, the results are not affected by the assumption. Of course, this is only true for the idealized conditions, assumed for the analytical model.

l 633f: What are the implications of this assumption for reconstructed wind vectors in a turbulence wind field?

The following explanation has been added to the manuscript (l 652):

"Under this assumption, all contributions to the measured radial velocity are attributed to the u-component of the wind field, which can lead to an overestimation of this component. The uncertainty related to this effect increases with growing magnitudes of v- and w- components. The implementation of other reconstruction approaches is possible but requires modifying the corresponding partial derivatives."

l 643: What is the measurement plane (the lidar measures along four lines)? Also, it depends on the spatial structure of the turbulent wind field if the samples taken along four beams are representative of the entire measurement plane.

We agree that the sentence might be misleading and believe it is not needed. Therefore, the sentence has been removed completely.

***Technical corrections***

All technical corrections have been implemented.

**Reply to RC-2**

The authors would like to thank the anonymous referee for the constructive and useful feedback. All comments have been taken into consideration for the revised version of the manuscript. A list of individual comments and replies follows:

General comments

A generally well-constructed and timely article dedicated to the quantification of motion induced uncertainties of wind speed derived from measurement of nacelle based lidar systems mounted on floating offshore wind turbines. The methodology is clear and explicit and given with enough details in such a way comparative investigation can be performed in an efficient way. There are, however, some assumptions that should be more elaborated.

The major result of the manuscript is that the uncertainty and bias in the 10 min wind speed are mainly driven by the pitch angle of the floater and the pitch amplitude. Thus, this paper presents a valuable contribution to the international wind energy community, since nacelle-based lidar measurements can be used to improve the speed regulation of wind turbines by a look-ahead update to the collective pitch control, thus reducing the loads applying on tower and blades and optimizing power output.

Globally the manuscript is easy to read, however, the authors should put some efforts on punctuation throughout the entire document, the missing of many commas make difficult to follow the authors' ideas. Also, the manuscript contains many typographical errors. I recommend a quick proofread. I've noticed that one reviewer makes the effort to provide an almost exhaustive list of the errors. It will help. Moreover, the manuscript will benefit from being rewritten in a more concise format. For these reasons, I cannot suggest the paper for submission in WES before some major revisions

The manuscript has been revised considering the general and specific comments of the reviewer. Overall, the manuscript has been shortened to 29 (31 first version) pages excluding Appendix and references.

**Specific comments**

*Abstract*

The abstract should be written in a more concise format and in a single paragraph.

The abstract has been revised according to the suggestion and written in a single paragraph.

*Introduction*

The literature is up to date. However, the main findings and weaknesses should be mentioned. What is lacking in these studies? What the authors couldn't or haven't done is their paper that you think it would worth being investigated? The objectives of your paper should be built to tackle what's missing in the cited papers.

The introduction has been revised, focusing on the suggestions and the similar comment of reviewer one.

Moreover, it is not clear, and not only in the introduction, on which metric the quantification of the uncertainties, the bias and the correction of motion is going to be applied. I finally figured it out that the wind speed is investigated after reading one third of the paper. It is often said that the analysis will be focus on "lidar measurement". It is not clear what's behind this expression. I had sometimes the feeling that turbulence, for example the turbulence intensity, would also be the topics of the paper. Expressions such as "probe volume averaging effect" (p. 7, line 180) have confused me. Please clarify.

The description of objectives has been revised to clarify this point. It is now explicitly mentioned that the metric investigated in the paper is wind speed (l 100).

p.4, line 99. It seems that surge and sway effects won't be addressed in the manuscript. It is then specifically written in page 5, line145 that both DOFs are not considered. Low-frequency fluctuations in surge and sway impact the LOS velocity fluctuations for floating lidar measurement. I agree that this low-frequency fluctuations are not governing the main dynamics of a nacelle, so both surge and sway can be put aside. However, it should be clarified in the text.

To avoid confusion about this point, the statement has been removed from line 99. In section 2.2 (Analytical model) a justification for the simplification in the analytical model has been included (l 156).

*Methodology*

The rotation matrix (Eq .7) should be written for the uninformed reader.

The rotation matrix has been added to the Appendix.

P11, line 243. Which method did you apply to compute the power spectral density? Welsh? Periodogram? Both methods are quite sensitive to the input parameters such as the "window". It should be clarified (method used and relevant input parameters). Hopefully, for your study the choice of the method wouldn't have a significant impact since you seek at identifying the spike induced by the motion. So please clarify briefly.

For this plot, the welch power spectral density estimate was used for the computation of the power spectral density with a hamming window dividing the time series into 32 segments.

*Results*

Since I've recommended to write a more concise version of the manuscript I'm wondering if figures 11, 12 and 15 and the associated text are really relevant. I think these parts could be removed or at least shorten without affecting the content of the paper.

As also suggested by reviewer one, figures 11, 12, 15 and associated text have been removed.

*Discussion*

I understand that the benefit of your analytical model is a gain in computational time but I'm wondering if it sufficient to chose this approach over more sophisticated approaches, i.e., model employing the generation of synthetic turbulent wind fields, to assess the uncertainty and bias in wind speed due to the movement? What would be your recommendations in term of model choice to a uniformed reader who would like to perform a similar study to yours? Specially what would be the impact of using a model with or without representation of turbulence?

The application of the analytical model requires less computational power. Additionally, the application has less requirements in terms of needed tools, skills and data. A first estimation of motion-induced effects can be made with basic design parameters of the floating turbine design without the need of aero-elastic simulations. In case it is found that motion-induced effects could be of relevance for the application of interest a simulation-based study is advisable. The comparison between the analytical and numerical models in our study showed little differences, indicating a small influence of turbulence. However, no sensitivity study, identifying the influence of individual parameters (TI, probe volume etc.), was conducted. We acknowledge that this should be addressed in future work.

The discussion has been revised, considering the discussion above (see l 559, 581)

**Reply to RC-3**

The authors would like to thank the referee for his constructive and useful feedback. All comments have been taken into consideration for the revised version of the manuscript. A list of individual comments and replies follows:

**General comment**

This paper presents a study of the measuring accuracy of a wind lidar mounted on the nacelle of a floating wind turbine. The authors perform an estimation of the expected uncertainties and biases on the wind lidar measurements that are introduced due to the motion of the floating wind turbine. I think that it is an overall well-written study and its topic is interesting for the readers of the Wind Energy Science journal. Please find below a list of minor comments and suggestions for improvements of the manuscript.

**Specific comments**

1. Line 157, Equation (3). I suggest using a different symbol for the translation velocities

   To avoid confusion about the used symbol, it is now explicitly stated that "x_vel" represents the vector of translational velocities in x,y,z.

2. Line 214: What is the function that describes f(a)? and is "a" the measurement distance or the distance from the measuring location?

   In this case "a" is the distance from the lidar to the focus point. As detailed in line 215 and 216, in the simulation environment, f is represented by the average over a definable number of points along the beam. This has been clarified in the manuscript.

   The notation has been slightly changed. The variable is now named a_d (equation 11).

3. Line 215: What is the length of range gate and how many points along the beam were used for the discretization?

   The length of the range gate is set to 30 m, discretized over 10 points. This information has been added to the manuscript ( l 240).

4. Line 245: Do the authors refer to Figure 4 (right) when they write: "the second peak is the tower natural bending frequency at around 1 Hz"? The plot presents the power spectral density down to 0.5 Hz. A second peak can be found around 0.3 Hz. Do the author mean that one?

   Accidentally, the explanation referred to an older version of the figure. This has been corrected.

5. Line 247: Is it the lidar pitch signal or the IMU pitch signal? If it is the former, on which basis this conclusion is drawn? If it the latter, isn't already stated in the sentences 241-242.

In fact, it is the lidar inclination sensor pitch signal. The section has been revised for clarity.

6. Line 265: Can you add which equation is used here to estimate the u- component?

The reconstruction approach is detailed in the Appendix. A reference to the Appendix has been added.

7. Line 269: Figure 5. Can you clarify in the manuscript which are the top and the which are the bottom beams? I guess that the VLos1 and VLos2 correspond to the top beams.

Indeed, VLOS1+2 correspond to the two upper beams, and 3,4 to the lower beams. The information has been added to the caption of the figure.

8. Line 284: The authors present in Figure 6 the impact that the translational velocities of the lidar device on the measured LOS velocities. The values of these velocities seem high. Have they been calculated based on the motion of the FLOATGEN demonstrator floating wind turbine.

The translational velocities, presented in this figure do not refer to any specific floater. The figure only aims to illustrate the impact of translational velocity on measured LOS velocities in general.

9. Line 345: The "lidar wind speed estimate" is the u-component estimate?

Correct, this fact has been clarified in the manuscript.

10. Lines 376-377: Why are biases up to 0.1 m/s (which are presented in Figure 9) characterized as significant?

We agree that it depends on the intended use of the data, to determine whether a level of bias is significant or not. For power performance measurements bias in the order of 0.1m/s or around 1% could be of importance. Therefore, it is considered as significant. The explanation has been added to the manuscript (l 411).

11. Line 385: The authors write that the "dynamics induced fluctuations in lidar wind speed measurements which are indicated by high measurements uncertainties". Why are these uncertainties considered high for a load evaluation?

We agree with the referee that the statement is not intuitively clear and believe that the quantification "high" is not necessary at this point. The statement has been changed to (l 419):

"It can be expected that the dynamics-induced bias in lidar wind speed measurements will affect load estimations."

12. Line 494: it is written "u-component wind speed of the original wind field averaged over the rotor plane". However, in line 496 it is written "the lidar measured u-component is observed". I guess that it is the same quantity in both cases. I think that this is confusing.

Therefore, I suggest that a different symbol is used to denote the longitudinal component of the wind vector from the rotor averaged wind speed throughout the manuscript.

These are two different quantities. One refers to the averaged u-component across all grid points of the wind field. The other one refers to the u-component reconstructed from the lidar measurements.

13. Line 502: In Figure 14 the power spectral density of the original wind field is presented along with uncorrected and filtered wind lidar lidar estimations. First, why is the maximum frequency 0.3 Hz? Shouldn't it be 0.5 Hz if the sampling rate of the lidar is 1 Hz? Also, the PSD of both the uncorrected and filtered lidar signals are higher than the original wind field in the frequency bandwidth from 0.075 Hz to 0.3 Hz. In line 509 it is stated that "the reference PSD of the original wind field represents an average over all points in the rotor plane. Therefore, the spectrum lies below the lidar estimated spectra". Why do the authors did not choose to plot the PSD of the reference u-component? The same question applies also to the results presented in Figure 16.

We agree, that the plot in figure 14 should be shown up to 0.5 Hz. The figure has been updated.
We considered showing the PSDs of the four lidar focus points instead of the average of all the points over the rotor plane. However, we believe that showing the PSD of the rotor average is more meaningful in this case since the rotor-averaged wind speed is used as reference throughout the paper. Please see the figure below for your reference.

[Figure]

Here, the violet line represents the PSD of the four grid points, corresponding to the lidar focus points.

**Technical corrections**

All technical corrections have been implemented

---

## Referee Report (RR1)

Please fix the typo in the last label in the legend of Figure 5 to "reconstructed".